# Genomic architecture and prediction of censored time-to-event phenotypes with a Bayesian genome-wide analysis

Sven E. Ojavee [1]✉, Athanasios Kousathanas [1], Daniel Trejo Banos [1], Etienne J. Orliac[2], Marion Patxot[1], Kristi Läll[3], Reedik Mägi[3], Krista Fischer[3,4], Zoltan Kutalik[5,6] & Matthew R. Robinson [7]✉

While recent advancements in computation and modelling have improved the analysis of complex traits, our understanding of the genetic basis of the time at symptom onset remains limited. Here, we develop a Bayesian approach (BayesW) that provides probabilistic inference of the genetic architecture of age-at-onset phenotypes in a sampling scheme that facilitates biobank-scale time-to-event analyses. We show in extensive simulation work the benefits BayesW provides in terms of number of discoveries, model performance and genomic prediction. In the UK Biobank, we find many thousands of common genomic regions underlying the age-at-onset of high blood pressure (HBP), cardiac disease (CAD), and type-2 diabetes (T2D), and for the genetic basis of onset reflecting the underlying genetic liability to disease. Age-at-menopause and age-at-menarche are also highly polygenic, but with higher variance contributed by low frequency variants. Genomic prediction into the Estonian Biobank data shows that BayesW gives higher prediction accuracy than other approaches.

[1] Department of Computational Biology, University of Lausanne, Lausanne, Switzerland. [2] Scientific Computing and Research Support Unit, University of Lausanne, Lausanne, Switzerland. [3] Estonian Genome Center, Institute of Genomics, University of Tartu, Tartu, Estonia. [4] Institute of Mathematics and Statistics, University of Tartu, Tartu, Estonia. [5] University Center for Primary Care and Public Health, Lausanne, Switzerland. [6] Swiss Institute of Bioinformatics, Lausanne, Switzerland. [7] Institute of Science and Technology Austria, Klosterneuburg, Austria. ✉email: svenerik.ojavee@unil.ch; matthew.robinson@ist.ac.at

G enome-wide association studies (GWAS) have greatly expanded our understanding of the genetic architecture of complex traits, but have largely focused on binary phenotypes and quantitative traits[1], leaving the age-at-onset of symptoms little studied, despite it being one of the key traits in biobank studies of age-related disease. Understanding the environmental and genetic basis of the time at which symptoms first occur is critical for early screening programs and for gaining insight into disease development and progression, especially as the pathological processes of many age-related diseases may be triggered decades before the first symptoms appear. Evidence suggests that genome-wide analyses conducted with case-control phenotypes tend to have less power in comparison with their age-at-onset analysis counterparts[2,3]. Genetic predictors created from case-control studies have been shown to be predictive of age-at-diagnosis[4], implying that early-onset is to a certain degree indicative of a higher underlying liability of disease. However, our understanding of the genetic architecture of reproductive timing, and the age at which symptoms first develop for common complex disorders, remains limited.

Statistical modelling of time-to-event data is a highly active research area and is frequently applied to clinical and pharmacogenetic studies. Analogous to single marker regression in GWAS analyses, a Cox proportional hazards (PH) model[5] for each single nucleotide polymorphism (SNP) $j \in \{1, \ldots, M\}$ can be formulated as $h_i(t) = h_0(t) \exp(\mathbf{x}_{ij}\boldsymbol{\beta}_j)$, where $h_0(t)$ is the baseline hazard at time $t$, $h_i(t)$ is the hazard for individual $i$, $\mathbf{x}_{ij}$ is the standardised $j$th SNP marker value, with $\boldsymbol{\beta}_j$ the effect size of the $j$th SNP and $M$ the total number of SNPs[6–8]. Recently there have been improvements in the computation times using some approximations for single-marker Cox PH regression[9], however, this approach still yields marginal effect size estimates as the markers are not fitted simultaneously. Residual based approaches have also been widely used, which first regress the phenotype on covariates such as gender or age at entry in Cox PH model, and then use the residuals in a second regression on the SNP data, with martingale residuals $\hat{M}_i = d_i - \hat{\Lambda}_0(t_i) \exp(Z_i^t \boldsymbol{\gamma})$, where $\hat{M}_i$ is the residual for individual $i$, $\delta_i$ is the failure indicator ($d_i = 1$ for the event during the study period, otherwise $d_i = 0$), $\hat{\Lambda}_0(t_i)$ is the baseline cumulative hazard function at time $t_i$, $t_i$ is the follow-up time for individual $i$, $Z_i$ is the vector of variables used in the first regression step and $\boldsymbol{\gamma}$ the vector of corresponding parameter estimates[10,11]. The martingale residual approach retains the linearity between the effect and the phenotype and given the model in the second step it can also be very fast. However, the failure time and censoring indicator are combined to one summary statistic, rather than including censoring information specifically via likelihood. Therefore, the martingale residual approach does not use the censoring information efficiently diminishing the power of this model. Rather than testing markers one-at-a-time, their effects can be estimated jointly in a mixed-effects Cox PH model, referred to as a frailty model, specified as $\lambda_i(t|b) = \lambda_0(t) \exp(X_i^t \boldsymbol{\beta} + b)$, where $\boldsymbol{\beta}$ is the effect for one SNP being tested along with other fixed effects such as age or sex, $b \sim N(0, \sigma^2 \Sigma)$ is the $N$-dimensional vector of random effects ($N$ is the sample size), $\Sigma: N \times N$ is the genetic relationship matrix, $\sigma^2$ is the variance of the genetic component, $\lambda_0(t)$ is the baseline hazard function and $\lambda_i(t|b)$ is the hazard for individual $i$. This idea has been long limited by computational resources and in the latest implementation (COXMEG)[12] analyses are constrained to around ~10,000 individuals. For joint marker effect estimation, there is also the Cox-LASSO model[13] which has been recently developed for genetic data in the R package snpnet[14,15]. Fully parametric alternatives are also the Sparse Bayesian Weibull regression (SBWR), which may outperform LASSO-based approaches[16], but like other Bayesian methods such as SurvEMVS[17] or a semi-parametric

g-prior approach of Held et al.[18] the ultrahigh dimensions of genetic data limit their application. Therefore, approaches that can efficiently handle both the complexity and scale of many millions of sequenced individuals with time-to-event outcomes have not been extensively developed, limiting our understanding and our ability to predict disease progression and the timing of symptom onset.

Here, we take an alternative approach to obtain accurate inference in full-scale phenotype-genotype sequence data sets, by proposing a mixture of regressions model with variable selection, using different regularisation parameters for genetically motivated groups (see "Methods" section). Our suggested model fits all of the markers jointly in a Bayesian framework using the Weibull assumption for the phenotypes. We show that this approach: (1) allows for a contrasting the genetic architectures of age-at-onset phenotypes under this flexible prior formulation; (2) yields marker effect estimates $\boldsymbol{\beta}_j$ that represent the effect of each marker conditional on the effects of all the other markers accounting for genetic architecture; (3) provides a determination of the probability that each marker and genomic region is associated with a phenotype, alongside the proportion of phenotypic variation contributed by each, and (4) gives a posterior predictive distribution for each individual. Regardless of the phenotypic distribution, our suggested approach greatly improves genomic prediction for the timing of events for each individual and enables better insight behind the genetic architecture underlying time-to-event traits.

## Results

**BayesW model.** An overview of our model is as follows, suppose that $M$ markers are split between $\Phi$ different groups. The groups can be for example formed based on marker-specific genomic annotations, MAF grouping, grouping based on LD score, etc. We assume for an individual $i$ that the age-at-onset of a disease $Y_i$ has Weibull distribution, with a reparametrisation of the model to represent the mean and the variance of the logarithm of the phenotype as

$$E(\log Y_i | \mu, \boldsymbol{\beta}, \boldsymbol{\delta}, \alpha) = \mu + \sum_{\varphi=1}^{\Phi} (\mathbf{x}_i^\varphi)' \boldsymbol{\beta}^\varphi + \mathbf{z}_i' \boldsymbol{\delta}, \qquad (1)$$

$$Var(\log Y_i | \mu, \boldsymbol{\beta}, \boldsymbol{\delta}, \alpha) = \frac{\pi^2}{6\alpha^2}, \qquad (2)$$

where $\mu$ is the intercept, $\mathbf{x}_i^\varphi$ are the standardised marker values for all SNPs in group $\varphi$, $\boldsymbol{\beta}^\varphi$ are the marker estimates for the corresponding group, $\mathbf{z}_i$ are additional covariate values (such as sex or genetic principal components), $\boldsymbol{\delta}$ is the additional covariate effect estimates and $\alpha$ is the Weibull shape parameter (see "Methods" section). For each group, we assume that $\boldsymbol{\beta}^\varphi$ is distributed according to a mixture of $L_\varphi$ Gaussian components. Each marker (from group $\varphi$) estimate $\boldsymbol{\beta}_j j \in \{1, \ldots, M\}$ is related to a corresponding indicator variable $\boldsymbol{\gamma}_j \in \{0, \ldots, L_\varphi\}$ where $L_\varphi$ is the number of mixture distributions. $\boldsymbol{\beta}_j$ have zero values if and only if $\boldsymbol{\gamma}_j = 0$. We assume that non-zero $\boldsymbol{\beta}_j$, where marker $j$ belongs to group $\varphi$, that has been assigned to mixture component ($\boldsymbol{\gamma}_j = k \geq 1$) comes from a normal distribution with zero mean and variance $C_k^\varphi \sigma_{G\varphi}^2$, that is $\boldsymbol{\beta}_j \sim N(0, C_k^\varphi \sigma_{G\varphi}^2)$, where $\sigma_{G\varphi}^2$ represents the phenotypic variance attributable to markers of group $\varphi$ and $C_k^\varphi$ is a group and mixture specific factor showing the magnitude of variance explained by this specific mixture that is given by the user. For example, specifying $C_1^\varphi = 0.0001$, $C_2^\varphi = 0.001$ and $C_3^\varphi = 0.01$ gives us mixtures that, respectively, explain 0.01%, 0.1% and 1% of the genetic variance. We also assume that prior probabilities of belonging to each of the mixture distribution $k$ is stored in $L_\varphi + 1$-dimensional vector $\boldsymbol{\pi}^\varphi$. Thus the mixture proportions,

variance explained by the SNP markers, and mixture constants are all unique and independent across SNP marker groups.

**An algorithm for whole-genome data at biobank scale**. We develop a computational framework that overcomes previous limitations for the application of age-at-onset models to large-scale biobank studies. In the model likelihood, we account for right censoring, a situation where only the last known time without an event is recorded, with the event potentially taking place sometime in the future (see "Methods" section). Although we did not apply it in our final analysis, we also formulate the model to accommodate left truncation, a situation where individuals are not missing from the data at random, creating differences in the genetic composition of individuals across age groups (see "Methods" section). We implement a parallel sampling scheme for Eq. (1) that allows the data to be split across compute nodes (in a series of MPI tasks), whilst still maintaining the accuracy of the estimation of $\boldsymbol{\beta}_j$. With $T$ parallel workers (MPI tasks), Bulk Synchronous Parallel (BSP) Gibbs sampling can sample $T$ marker effects when sequential Gibbs samples a single one, but BSP requires an extra synchronisation step of the tasks after each of them has processed $u$ markers (see "Methods" section). After each worker has processed $u$ markers, we synchronise the workers by transmitting the residual vector across workers. Given our assumption that the phenotype follows a Weibull distribution, we are using a numerical method, Adaptive Gauss-Hermite quadrature, for calculating the mixture membership probabilities for variable selection, and Adaptive Rejection Sampling (ARS) for estimating the marker effects. We implement these approaches to take full advantage of the sparsity of genomic data, converting computationally intensive calculations of exponents and dot products into a series of summations. We provide publicly available software (see "Code Availability") that has the capacity to easily extend to a wider range of models (not just Weibull) than that described here. Our software enables the estimation of 2,975,268 SNP inclusion probabilities split between $T = 96$ workers, using 12 compute nodes and synchronisation rate of $u = 10$, mixture allocation and effect sizes in 371,878 individuals with an average of 49.7 s per iteration without groups model and 50.0 s per iteration with the groups model; using 151,472 individuals with $T = 64$, 8 compute nodes and $u = 10$ without groups we get an average of 24.8 and with groups, we get an average of 26.4 s per iteration. Here, we have chosen to run the chains for 10,000 iterations leading to execution times of 69 h ($N = 151,472$) to 139 h ($N = 371,878$). The run times ultimately depend upon compute cluster utilisation and the genetic architecture of the phenotype, as calls to the ARS procedure are linear with the number of markers. The calculations were done by using Helvetios cluster of EPFL (see Code availability).

**Simulation study**. We show in a simulation study that our model estimates SNP marker effect sizes more accurately, with a greater number of discoveries, and thus obtains better model performance with improved genomic prediction accuracy as compared to other available methods (Fig. 1, Supplementary Figs. 1 and 2). The other methods used for comparisons in simulations are Cox-LASSO[13], Bayesian regression mixture model BayesR[19] applied on martingale residuals and marginal single marker regression (OLS) applied on martingale residuals. First, we show that the previous statement holds even in the case of model misspecification (Fig. 1), where the phenotypic distribution does not correspond to a Weibull distribution, but rather conforms to a series of different generalised gamma distributions (of which Weibull is one of them in the case where $\theta = 1$ in the parametrisation of Eq. (1) in the Supplementary

Note of Supplementary Information), with differing $\theta$ value (see "Methods" section and Supplementary Note in Supplementary Information). In a simulation study of $N = 5000$ individuals and 50,000 uncorrelated markers with $p = 500$ randomly selected markers as causal variants, our BayesW model obtains higher out-of-sample prediction accuracy than a Cox-LASSO or a martingale residual approach used in several recent studies (Fig. 1a).

Second, we show that this statement also holds in a larger simulation study using a real genomic data-set of $N = 20,000$ randomly selected UK Biobank individuals and 194,922 correlated genetic markers on chromosome 22 under different censoring levels (Supplementary Fig. 1). Interestingly, in Fig. 1a we observe that the generalised gamma distributions with $\theta > 1$ lead to more accurate genetic predictions compared to the Weibull model ($\theta = 1$). Such phenotypic distributions are easier to discriminate meaning that for distributions where $\theta > 1$ the same difference in genetic values leads to greater phenotypic distribution differences in Kullback-Leibler divergences compared to $\theta = 1$. Our approach achieves better precision-recall as compared to these approaches (Fig. 1b) across all values of $\theta$ and all censoring levels within the data (Supplementary Fig. 2). We choose to use precision-recall curves due to the great imbalance between the number of causal and non-causal markers[20]; precision ($\frac{\text{TP}}{\text{TP+FP}} = 1 - \text{FDR}$) describes how accurately the markers were identified while recall ($\frac{\text{TP}}{\text{TP+FN}}$) describes the proportion of how many causal markers were discovered. We show that across the range of $\theta$ values that generate model misspecification, SNP marker effect estimates remain mostly correctly estimated (Fig. 1c, Supplementary Fig. 9), however, due to the shrinkage effect of the prior distribution to the marker effect size estimates, we observe a very slight underestimation of the effect size estimates for this simulation scenario if the model is correctly specified. On the other hand, if the phenotype is from log-normal distribution ($\theta = 0$) then due to the inflated genetic variance hyperparameter (Fig. 1d) we see the reduced impact of the priors and less shrinkage of the effect size estimates, leading to more accurate effect size estimates. In general, we recognise that Bayesian modelling may induce slight shrinkage in the effect size estimates due to the priors. Nevertheless, we consider this effect negligible (Supplementary Fig. 9), especially in the context of improved genetic prediction and a more flexible framework that Bayesian modelling enables.

Third, in the Supplementary Note, we derive a definition of SNP heritability, the proportion of among-individual variation in age-at-onset that is attributable to SNP effects, for both the variance of the logarithmed phenotype and the original scale (see Supplementary Note). We show that the log-scale SNP heritability definition is valid under a Weibull assumption and across the range of theta when restricting the markers entering the model (Fig. 1d, single mixtures 0.01), but maybe inflated under low theta values (Fig. 1d, mixtures 0.001, 0.01) because of the increase in small-effect false positives that enter the model (Fig. 1e). In addition, we demonstrate that the model is robust to the specification of mixture components (Fig. 1f), false discovery rate is bounded even if we add smaller mixtures. To explore false discovery rate and polygenicity in a more realistic scenario we further simulate different numbers of causal loci on LD pruned set of UK Biobank chromosome 1 ($M = 230,227$) (see "Methods" section). We show that (a) our model captures accurately the effect size distribution (Supplementary Fig. 11a), (b) our model accurately captures the underlying polygenicity (Supplementary Fig. 11b), (c) our model controls for false discovery rate (Supplementary Fig. 12). Throughout this work, we use the posterior probability of window variance (PPWV)[21]

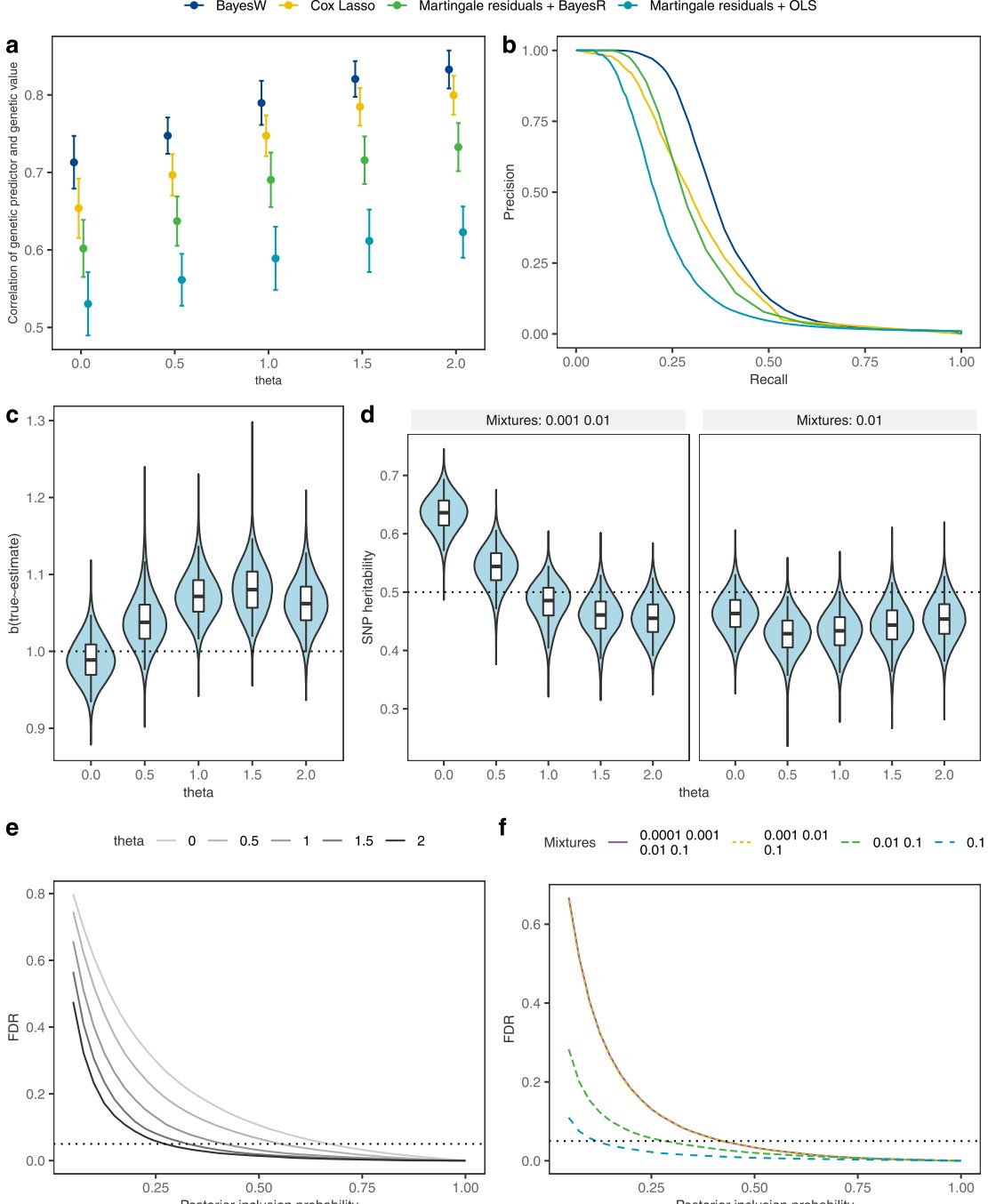

**Fig. 1 Simulation study results.** Method comparison, parameter estimation results and the behaviour of the false discovery rate (FDR). Models were estimated on a data set of $M = 50{,}000$ uncorrelated markers and $N = 5000$ individuals in 25 replicate simulations of 5 chains with 3000 iterations. Phenotypes were created from Generalised gamma distributions (see Supplementary note) using $p = 500$ causals markers and retaining heritability of $h^2 = 0.5$; independent data set had the same markers with $N = 1000$ other individuals. **a** Prediction accuracy of four methods when predicting to an independent data set given different generalised gamma distributions. The plot centres indicate the mean and error bars indicate the standard deviation of the correlations across simulations; (**b**) Mean precision values for each level of recall for four methods using Weibull phenotype (theta = 1); (**c**) Regression slope (true effect size ~ estimated effect size) when estimating non-zero marker effects given different theta values estimated with BayesW at each iteration across all simulations; (**d**) BayesW SNP heritability estimates given different generalised gamma distributions and different used mixtures at each iteration across all simulations; (**e**) relationship between the posterior inclusion probability (PIP) and false discovery rate (FDR) given different generalised gamma distributions for BayesW, for each PIP we present mean FDR values; (**f**) relationship between the PIP and FDR for a different number of mixture distributions used using Weibull phenotype and BayesW, for each PIP we present mean FDR values. In panels **c**–**d**, the bounds of the box show the interquartile range, centre shows the median and minimum and maximum indicate the 95% credibility interval.

(see "Methods" section) as the metric to summarise the significance of genetic regions. PPWV shows the probability that a genetic region explains at least some fixed proportion of the genetic variance. We show below that false positives are controlled for under all generative models when conducting LD clumped based variable selection using a PPWV threshold of ≥0.9 (Supplementary Fig. 12) and hence it is justified to use it for calling region-based discoveries and compare it with other methods that are supposed to control for false discovery rates. Finally, we show that our BSP algorithm is stable under a wide range of synchronisation rates, parallelism, and quadrature point selection (Supplementary Fig. 3).

**The genetic architecture of age-at-onset**. We then applied our model to unrelated UK biobank individuals of European ancestry with a pruned set of $M = 2,975,268$ SNPs for five traits: two reproductive phenotypes of age-at-menopause ($N = 151,472$) and age-at-menarche ($N = 200,493$) and three common complex diseases (selected as they are some of the leading causes of mortality) of time-to-diagnosis of type-2-diabetes (T2D) ($N = 372,280$), coronary-artery-disease (CAD) ($N = 360,715$) and high blood pressure (HBP) ($N = 371,878$) (see Descriptive statistics in Supplementary Table 1). Using our BSP Gibbs sampling scheme, we ran a baseline model without any grouping of markers, and then we re-ran the model grouping markers into 20 MAF-LD bins (quintiles of MAF and then quartiles within each MAF quantile split by LD score). Groups were defined using MAF and LD based on recent theory[22] and recent simulation study results[23–26], which suggest that accurate estimation of genetic variance might require accounting for the MAF-LD structure. To understand the effect size distribution and genetic architecture, four mixture components were specified such that they would represent 0.001%, 0.01%, 0.1% or 1% of the total genetic variance for the no groups model (0.00001, 0.0001, 0.001, 0.01). For the group model, the four group-specific mixtures for each of the 20 groups were chosen to be 10 times larger (0.0001, 0.001, 0.01, 0.1) such that they would represent 0.01%, 0.1%, 1% or 10% of the group-specific genetic variances. Additional variables such as sex, UK Biobank assessment centre, genotype chip, and the leading 20 PCs of the SNP data (see "Methods" section) were used as fixed effects in the analysis. We conducted a series of convergence diagnostic analyses of the posterior distributions to ensure we obtained estimates from a converged set of chains (Supplementary Figs. 4, 5, 6 and 7).

Under the assumption that the traits are Weibull distributed, this gives log-scale SNP (pseudo-)heritability estimates (see Supplementary Note of Supplementary Information) of 0.26 (95% CI 0.25, 0.27) for age-at-menopause, 0.41 (95% CI 0.40, 0.42) for age-at-menarche, 0.36 (95% CI 0.35, 0.37) for age-at-diagnosis of HBP, 0.48 (95% CI 0.44, 0.52) for age-at-diagnosis of CAD, and 0.52 (95% CI 0.50, 0.55) for age-at-diagnosis of T2D. Both the model with and without groups reach similar conclusions in terms of partitioning markers between mixtures (Fig. 2a) indicating that the inference we draw on the genetic architecture is here independent of the group-specific prior specification. However, our BayesW grouped mixture of regression model allows for contrasting the variance contributed by different MAF and LD groups across traits. For all traits, we find that the majority of the variance contributed by SNP markers is attributable to SNPs that each proportionally contribute an average of $10^{-5}$ of the genetic variance (Fig. 2a). We find evidence that age-at-menarche is highly polygenic with 88.1% (95% CI 86.8%, 89.4%) of the genetic variance attributable to the SNPs contributed by markers in the $10^{-5}$ mixture group, similar to CAD with 74.2% (95% CI 63.6%,

81.5%, Fig. 2b). Age-at-menopause and age-at-T2D diagnosis stand out with 32.3% (95% CI 28.9%, 35.7%) and 18.9% (95% CI 14.6%, 22.9%) of the genotypic variance attributable to the SNPs contributed by markers in the $10^{-3}$ mixture, respectively (Fig. 2b), indicating a substantial amount of genetic variance resulting from moderate to large effect sizes. In contrast, for the other traits, the moderate to large effect sizes (mixture $10^{-3}$) explain a far smaller part of the total genetic variance with age-at-menarche having almost no genetic variance (0.1%, 95% CI 0.0%, 0.6%) and only a small amount coming from that mixture for age-at-HBP diagnosis (5.6%, 95% CI 3.1%, 8.4%) and age-at-CAD (9.4%, 95% CI 6.5%, 12.9%).

We find marked differences in the underlying genetic architecture of these different age-at-onset phenotypes (Fig. 2c, d). For age-at-menarche, many rare low-LD SNPs and many common SNPs contribute similar proportions to the phenotypic variance attributable to the SNP markers, implying larger absolute effect sizes for rare low-LD variants per minor allele substitution, with age-at-menopause showing a similar but less pronounced pattern with a noticeable proportion of the genetic variance stemming from small effect sizes of the rare variants (Fig. 2d, MAF quintiles 1–3). In contrast, we find evidence that the phenotypic variance attributable to the SNP markers for age-at-diagnosis for CAD, HBP, and T2D is predominantly contributed by common variants of small effect (Fig. 2d). This implies that female reproductive traits may have been under far stronger selection in our evolutionary past than age-at-diagnosis of modern-day common complex disease[27]. In summary, we find that most of the phenotypic variance attributable to SNPs is contributed by very many small-effect common variants, but that there are key differences among time-to-event phenotypes, with reproductive traits showing different patterns of genetic architecture to time-to-diagnosis phenotypes.

We then partitioned the SNP markers into regions of LD clumps (see "Methods" section) and determined the genetic variance each of those regions explains. Then, we calculated the probability (PPWV) that each such region contributes at least 1/1000, 1/10,000 or 1/100,000 of the total genotypic variance, providing a probabilistic approach to assess the contribution of different genomic regions to time-to-event phenotypes. The smallest threshold was chosen to be 1/100,000 of the total genotypic variance corresponding to the smallest mixture component models were estimated with which also represents the magnitude of the smallest effect size we intend the model to capture. We find 291 LD clumped regions for age-at-menarche with ≥0.95 PPWV of 1/100,000, 176 regions for age-at-menopause, 441 regions for age-at-diagnosis of HBP, 67 regions for age-at-diagnosis of CAD, and 108 regions for age-at-diagnosis of T2D from our BayesW grouped mixture of the regression model (Fig. 3a). Our grouped model provides slightly better model performance, as reflected by higher posterior inclusion probabilities at smaller effect sizes (Fig. 3a, Supplementary Fig. 8), with the baseline BayesW mixture of regression model detecting 13.7% fewer LD clump regions for age-at-menarche, 4.5% fewer for age-at-menopause, 34.0% fewer for age-at-diagnosis of HBP, 35.8% fewer for age-at-diagnosis of CAD, and 33.3% fewer for age-at-diagnosis of T2D when using 1/100,000 PPWV threshold. Similarly, we evaluated region-based significance by calculating PPWV for regions that were created by mapping markers to the closest gene (Fig. 3b). For age-at-menopause we find 101, for age-at-menarche, we find 119, for time-to-T2D we find 41, for time-to-HBP we find 159 and for time-to-CAD, we find 20 gene-associated regions with ≥95% PPWV of explaining at least 1/10,000 of the genetic variance. In addition, we find evidence for differences in the effect size distribution across traits, largely reflecting differences in power that result from sample size

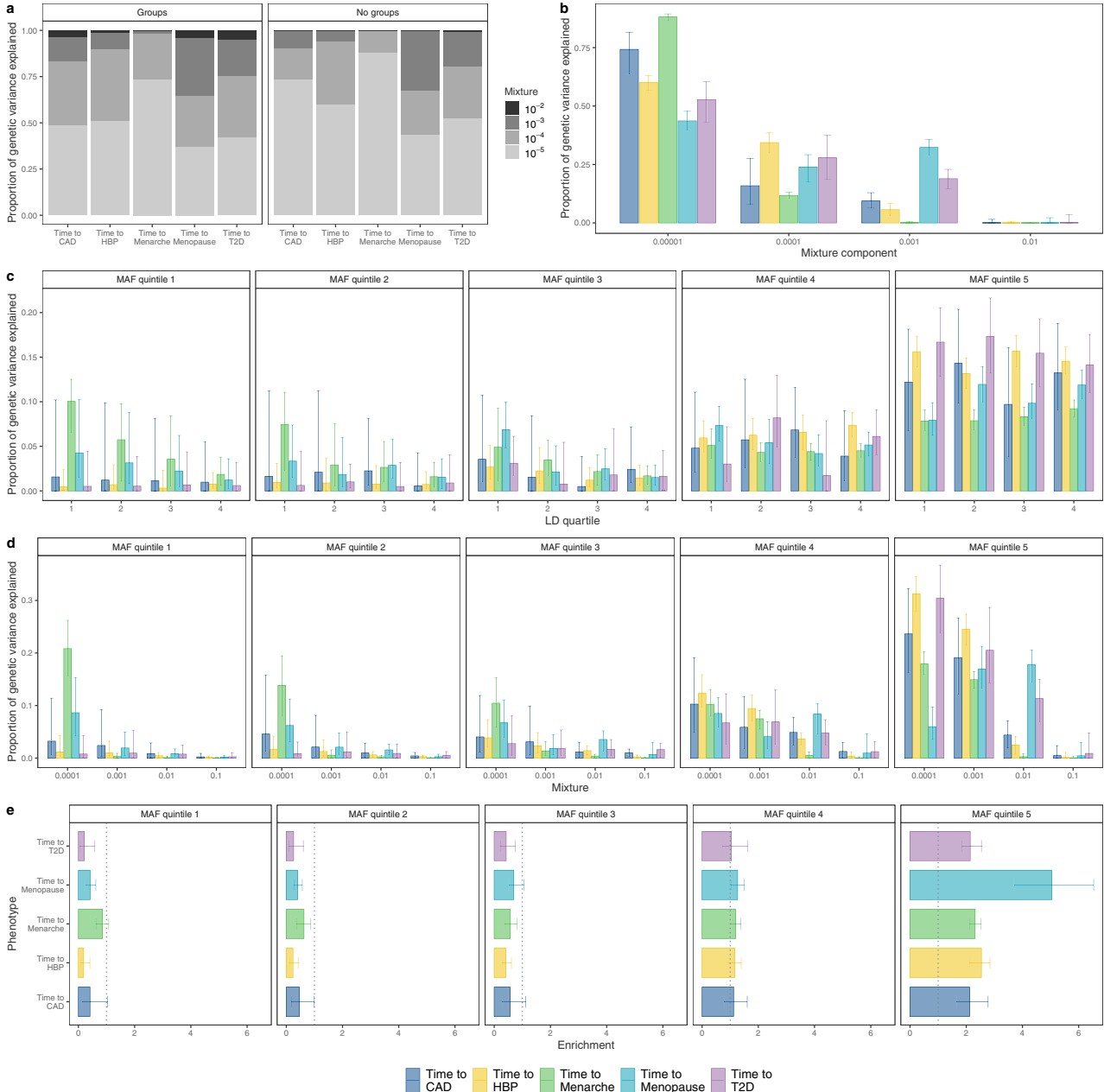

**Fig. 2 Genetic architecture for time-to-diagnosis of CAD, HBP, T2D and age-at-menarche and age-at-menopause, estimated using 2,975,268 markers and unrelated European ancestry individuals from the UK biobank.** BayesW models were executed with and without partitioning the markers into groups. Groups were created by splitting them first into MAF quintiles and each MAF quintile was further split into quartiles based on LD giving us a total of 20 groups. The used sample sizes were $N = 360,715$; 371,878; 372,280; 200,493 and 151,472 for time-to-diagnosis of CAD, HBP, T2D and age-at-menarche and age-at-menopause, respectively. The models without groups were run with five chains and the models with groups were run with three chains, each chain contained 10,000 iterations out of which 5000 first iterations were discarded as burn-in and a thinning step of five was applied to give 1000 samples from each chain. **a** Mean proportions of genetic variances explained by each of the mixtures with the groups model and without groups model, groups model and no group model are yielding rather similar results; (**b**) distribution of proportion of genetic variance between mixtures for the model without groups, time-to-menarche stands out with almost all of genetic the variance attributed to the small mixtures; (**c**) distribution of proportion of genetic variance between LD quartiles within each MAF quintile, LD bins do not have a large impact on genetic variance partitioning as the credibility intervals are large and medians across LD quartiles are rather stable; (**d**) distribution of proportion of genetic variance between mixtures within each MAF quintile, mixture allocations tend to be similar compared to no groups model; (**e**) enrichment (ratio of proportion of genetic variance and proportion of markers attributed to each MAF quintile group) value for each phenotype, enrichment of higher than 1 represents that the markers are explaining more of the genetic variance compared to their count proportion and vice versa. **b**–**e** Height of the bars represents median and error bars represent 95% credibility intervals, (**c**–**e**) are group models. For all of the traits, most of the genetic variance is coming from common SNPs (MAF quintile 5).

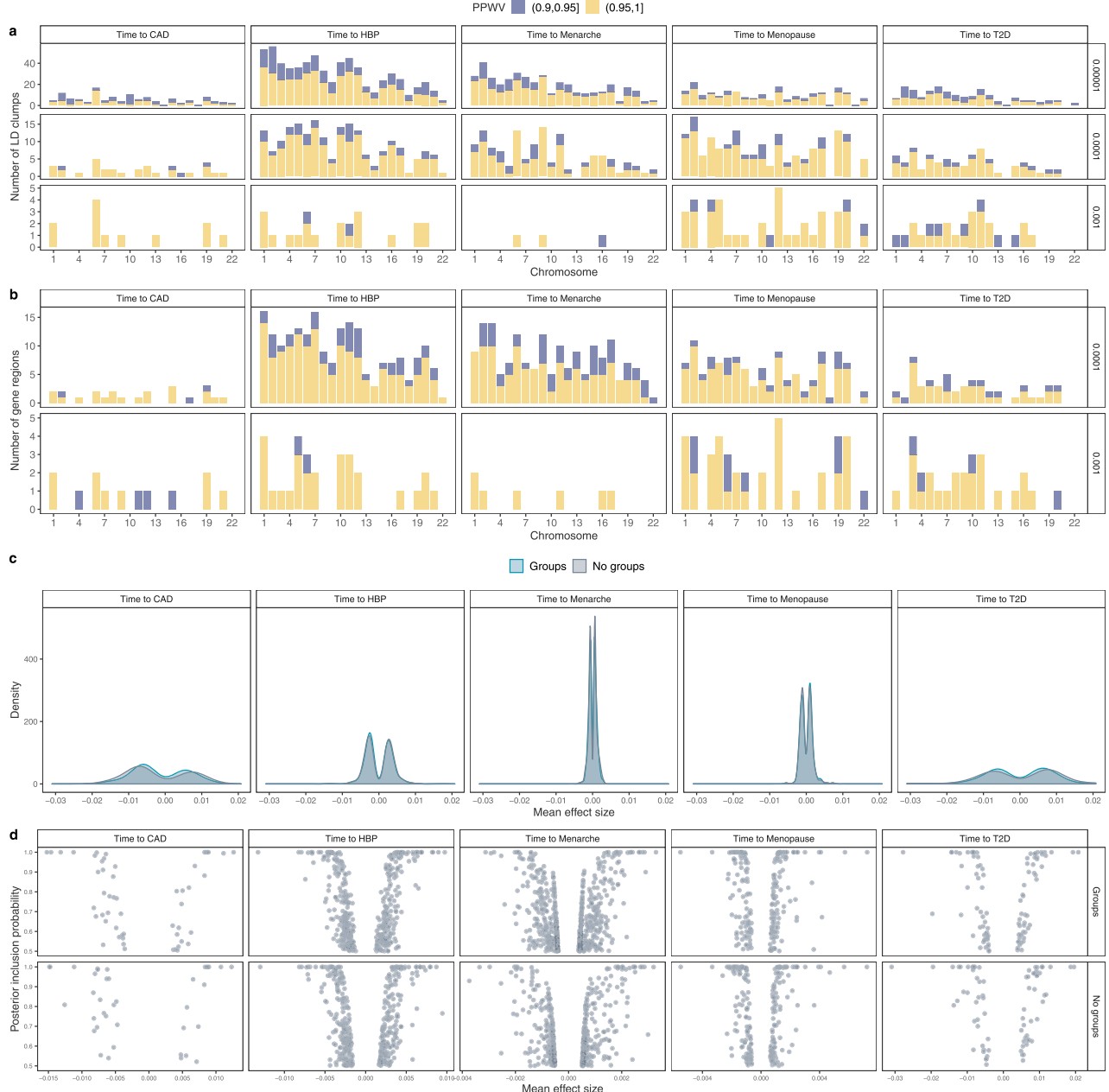

**Fig. 3 Regional and individual SNP contributions to the time-to-diagnosis of CAD, HBP, T2D and age-at-menarche and age-at-menopause. a** Count of LD clumped regions with high PPWV (Posterior Probability Window Variance). We split the genome into LD clumped regions such that the $r^2 < 0.1$ between index SNPs. Then we calculated the probability that a region is explaining at least either 0.001%, 0.01% or 0.1% of the genetic variance (PPWV); (**b**) Count of gene-mapped regions with high PPWV. Each marker is mapped to the closest gene given that it is within ±50 kb from the marker, then for each of those gene-specific regions, we calculate PPWV that the region explains either 0.01% or 0.1% of the genetic variance. Both **a** and **b** are using the groups model; (**c**) distribution of mean non-zero effect sizes for markers with PIP > 0.5 for models with and without groups, we pick up most large effects for traits such as time-to-diagnosis of CAD or T2D whereas we find an abundance of small effects for age-at-menarche, we see a very small effect of penalisation in the case of group model; (**d**) relationship between mean non-zero effect size and posterior inclusion probability for markers with PIP > 0.5 with markers. Source data are provided as a Source Data file.

differences and different censoring levels across traits (Fig. 3c, d) (see "Methods" section). Overall, these results suggest many hundreds of genomic regions spread throughout the genome contribute to the timing of common complex traits.

We also compared the LD clumped regions discovered by BayesW with LD clumped regions discovered by another association method fastGWA[28]. Even though fastGWA is a frequentist and BayesW is a Bayesian method and the comparison between the two approaches is not comprehensive,

we still use it as both methods should control for false discovery rate (Supplementary Fig. 12) and fastGWA is one of the most recent methods released. Although for age-at-menarche and age-at-menopause we find 191 and 97 regions that are concordantly significant according to the two methods (Table 1), we find less concordance among the other traits. For time-to-angina and -heart attack fastGWA does not find any significant regions, for time-to-HBP BayesW finds greatly more LD clumped regions (BayesW: 663, fastGWA: 14). The striking difference between the

**Table 1 Concordance between the LD clumped regions discovered by BayesW or fastGWA.**

| Phenotype | BayesW: no fastGWA: no | BayesW: no fastGWA: yes | BayesW: yes fastGWA: no | BayesW: yes fastGWA: yes | Total BayesW |
|---|---|---|---|---|---|
| Time to Angina | 290,220 | 0 | 128 | 0 | 128 |
| Time to Heart attack | 289,787 | 0 | 128 | 0 | 128 |
| Time to HBP | 291,674 | 4 | 653 | 10 | 666 |
| Time to Menarche | 292,127 | 242 | 223 | 191 | 414 |
| Time to Menopause | 292,202 | 125 | 126 | 97 | 227 |
| Time to Diabetes | 290,599 | 40 | 174 | 8 | 183 |

We split the genome into LD clumped regions and we evaluated the significance of each of the regions using the results from the groups BayesW model and the fastGWA model. The fastGWA results for our CAD and T2D definition were missing so instead time-to-angina and time-to-heart attack are shown for CAD and time-to-diabetes is shown for T2D. Here, BayesW calls an LD clumped region significant if the PPWV of the region (explaining at least 0.001% of the genetic variance) is higher than 0.9; fastGWA calls an LD clumped region significant if there exists at least one marker with a p-value $< 5 \times 10^{-8}$. We find that although for age-at-menarche and age-at-menopause there exists an abundance of regions with concordant significance, for other traits most of the discovered regions differ between two methods. For creating the comparison only overlapping markers were used; in the column Total BayesW we show the total number of discovered LD clumped regions, including those that did not have a counterpart among fastGWA results.

numbers of identified regions could be largely attributed to the larger sample size of BayesW as BayesW can also use the data from censored individuals where fastGWA can only resort to the uncensored individuals. For time-to-diabetes BayesW identifies more than three times more regions but only a small minority of the discovered regions are concordant. Although for time-to-menarche the fastGWA identifies more LD clumped regions, still around half of the regions identified by BayesW are not picked up by fastGWA. We further looked into the properties behind the discoveries that are not concordant between the two methods. It can be seen that the regions discovered by BayesW have lower p-values compared to the overall p-values (Supplementary Fig. 13a) indicating that many of those regions could be lacking power with fastGWA whereas BayesW manages to identify them; similarly, the regions that are discovered by fastGWA and not by BayesW tend to have higher PPWV compared to the overall PPWV values (Supplementary Fig. 13b) indicating that some potential signal could be lost when using such PPWV threshold. In terms of the prediction accuracy, the BayesW shows greatly better prediction accuracy to Estonian Biobank compared to fastGWA when predicting age-at-menarche or age-at-menopause (Fig. 4a, b) indicating that the regions identified by BayesW and their effect size estimates might reflect the genetic architecture more accurately. Therefore, BayesW identifies already found regions along with previously unidentified regions compared to previous association methods; for time-to-diagnosis traits, it can discover more regions due to using the censored individuals; and BayesW results yield greatly improved prediction accuracy compared to fastGWA.

**Out of sample prediction in an Estonian population.** We used the estimates obtained from the group-specific model to predict time-to-event in $N = 32,594$ individuals of the Estonian Biobank data. We compared our model performance to the Cox-LASSO approach implemented in the R package snpnet[14,15] trained on the same UK Biobank data (see "Methods" section) using two metrics. As some of the Estonian Biobank time-to-event phenotypes are censored, we choose to calculate the $R^2$ values between the predicted values and the martingale residuals from the Cox PH model where the true phenotypes are regressed on sex. In addition, we calculate Harrell's C-statistic[29] from the Cox PH model where the true phenotypic values are regressed on the predicted values and sex. BayesW outperforms Cox-LASSO for all phenotypes (Fig. 4a,b) by giving $R^2$ of 0.032 compared to Cox-LASSO's 0.017 for age-at-menopause of 18,134 women and 0.05 compared to Cox-LASSO's 0.040 for age-at-menarche of 18,134 women. We also get an increase in Harrell's C-statistic with BayesW giving 0.623 (se = 0.00443) compared to Cox-LASSO's

0.593 (0.00455) for age-at-menopause and for age-at-menarche we get C-statistic of 0.598 (0.00290) with BayesW compared to Cox-LASSO's 0.580 (0.00294). For the age-at-diagnosis traits, we obtain $R^2$ values of 0.0047, 0.0236, and 0.0441 for BayesW and 0.0030, 0.0135 and 0.0271 with Cox-LASSO for CAD, T2D, and HBP, respectively. This shows that our BayesW model gives a higher prediction accuracy compared to the Cox-LASSO method, in-line with our simulation study results.

We then compared the BayesW prediction results to those obtained from a case-control analysis. In a companion paper[22], we develop a group-specific BayesR approach and we use this to analyse the indicator variable (0 = no registered disease, 1 = reported disease) for HBP using the same data and a liability model to facilitate a direct comparison of the methods. For CAD and T2D, we use the results of the companion paper, where there were almost twice as many case observations (for CAD BayesR had 39,776 vs. BayesW 17,452 and for T2D BayesR had 25,773 vs. BayesW 15,813 cases) as it included those with the confirmed diagnosis but no age information and 8.4 million SNPs were analysed. For the prediction of age-at-diagnosis, we compared the $R^2$ values between the predicted values and the martingale residuals from the Cox PH model and the Harrell's C-statistic (Fig. 4a,b). For HBP, CAD we find that BayesW marginally outperforms BayesR with (HBP $R^2$ BayesW 0.0441, BayesR 0.0437; CAD $R^2$ BayesW 0.0047, BayesR 0.0046) and for T2D BayesR marginally outperforms BayesW ($R^2$ BayesW 0.0236, BayesR 0.0262). A similar ranking can be observed when using Harrell's C-statistic for comparison (Fig. 4b). We then compared approaches when predicting 0/1 case-control status, rather than age-at-diagnosis (Fig. 4c, d). We find that for predicting HBP BayesW marginally outperforms BayesR in terms of $R^2$ (BayesW 0.0375, BayesR 0.0365) and area under PR curve (BayesW 0.339, BayesR 0.336) (used because of the imbalance between cases and controls); for predicting CAD or T2D despite the increase case sample size, BayesR only marginally outperforms BayesW (T2D $R^2$ BayesW 0.0127, BayesR 0.0136 and AUC BayesW 0.0766, BayesR 0.0799; CAD $R^2$ BayesW 0.0025, BayesR 0.0027 and AUC BayesW 0.0920, BayesR 0.0941). Therefore we get very similar prediction accuracies with both methods when predicting case-control phenotypes although the BayesW model was estimated using time-to-event phenotypes with fewer cases for T2D and CAD.

A finding of similar prediction accuracy is unsurprising given the striking concordance between the results of the two models when partitioning the genotype into 50 kb regions. We calculated (on a logarithmic scale) the mean proportion of genetic variance attributed to each 50kb region for the model using case-control phenotype (BayesR) and for the model using time-to-event phenotype (BayesW). Both models attribute similar amounts of

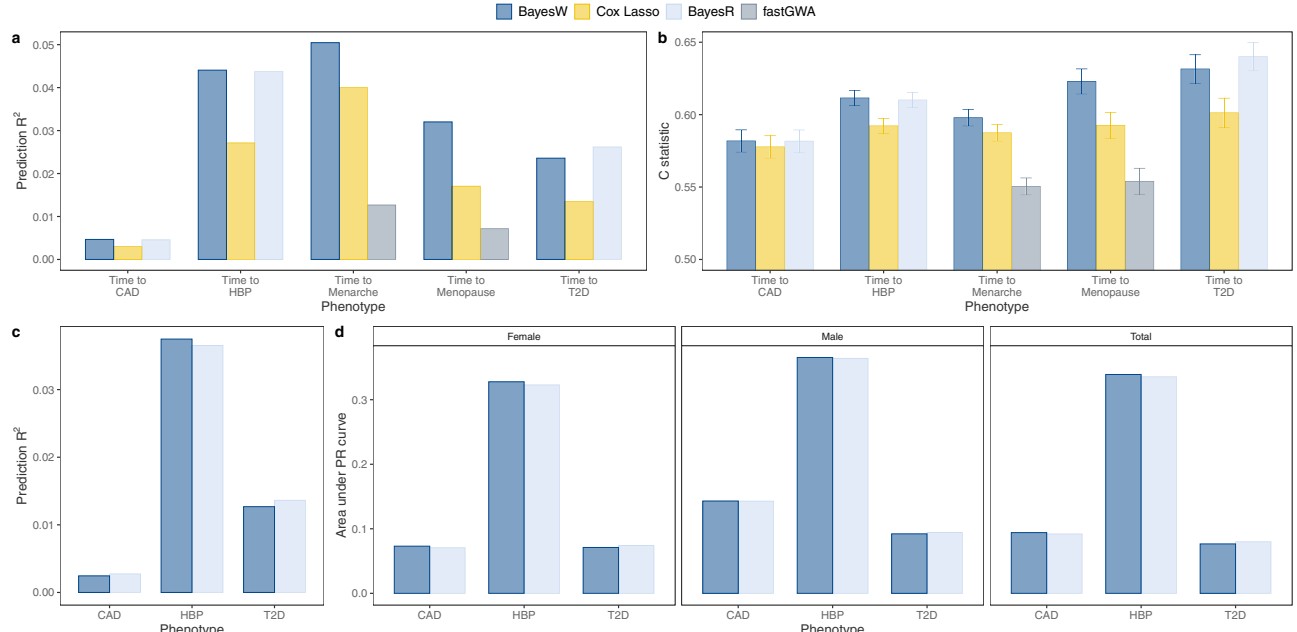

**Fig. 4 Prediction into the Estonian biobank.** BayesW and Cox-LASSO (estimated with snpnet) methods were used for all of the phenotypes; BayesR was used to see how case-control model predicts time-to-diagnosis phenotypes (CAD, HBP, T2D); fastGWA was used to see how marginal model performs when predicting continuous traits. BayesW and Cox-LASSO models were estimated using sample sizes of $N = 360{,}715$; 371,878; 372,280; 200,493 and 151,472 for time-to-diagnosis of CAD, HBP, T2D and age-at-menarche and age-at-menopause, respectively. The number of validation individuals from the Estonian Biobank was $N = 32{,}594$ for the time-to-diagnosis traits and $N = 18{,}134$ and 19,368 for age-at-menarche and age-at-menopause, respectively. **a** Prediction $R^2$ when predicting Estonian martingale residuals of time-to-event phenotypes using either BayesW, Cox-LASSO, BayesR or fastGWA model trained on the UK biobank data, martingale residuals were calculated from Cox PH model with an intercept and sex if applicable; (**b**) Harrell's C-statistic, calculated from Cox PH model where true phenotype was regressed on the genomic prediction and gender data (for CAD, HBP and T2D). Height of the bars represents the C-statistic and error bars represent corresponding 95% confidence intervals; (**c**) Prediction $R^2$ when predicting Estonian binary phenotypes (CAD, HBP, T2D) using either a model based on time-to-event data (BayesW) or case-control data (BayesR), the binary phenotypes that were used for the comparison were adjusted for age and sex; (**d**) Area under Precision-recall curve when predicting Estonian binary phenotypes (CAD, HBP, T2D) using either BayesW or BayesR, areas under the curve were calculated separately for females, males and everyone combined.

genetic variance to the same 50 kb regions (Supplementary Fig. 10), with correlations between logarithmic proportions of genetic variances are 0.941, 0.647 and 0.554 for HBP, T2D and CAD, respectively. Thus, we have shown here that using either time-to-event or case-control data for genome-wide association analysis we find a similar amount of genetic variance attributed to the same regions and both analyses have similar predictive performance when predicting case-control phenotypes. This suggests that to some extent there is interchangeability between the case-control and time-to-event phenotypes demonstrating that both phenotypes are describing a similar latent mechanism.

The BayesW model enables posterior predictive distributions to be generated for each individual. For evaluating the predictive performance of BayesW on the reproductive traits, we calculated the 95% credibility prediction intervals for each of the subjects in the Estonian Biobank. We chose to evaluate reproductive traits only as it is sure that every woman should experience those events given they have reached a sufficient age. For age-at-menopause 92.3% and for age-at-menarche 94.8% of the true uncensored phenotypes lie within the BayesW 95% credibility prediction intervals. This demonstrates that even though the prediction $R^2$ values for those traits are not very high due to the low genetic variance underlying the phenotypic variance, our Bayesian model quantifies the model uncertainty and yields well-calibrated subject-specific prediction intervals. An example of the shape of the distribution is shown in Supplementary Fig. 15. A caveat is that the subject-specific prediction intervals can be rather wide, though approximately half the width of the data range. For age-at-menopause, the data range from 34 to 63

(width of 29 years), and the width of the 95% credibility intervals ranged from 13.4 to 18.6 years with a median width of 15.9 years. For age-at-menarche, data values range from 9 to 19, and the posterior predictive interval width ranged from 5.1 to 7.9 years with a median width of 6.3 years.

## Discussion

Here, we have shown that our BayesW mixture of regressions model provides inference as to the genetic architecture of reproductive timing and the age at which symptoms first develop for common complex disorders. We provide evidence for an infinitesimal contribution of many thousands of common genomic regions to variation in the onset of common complex disorders and for the genetic basis of age-at-onset reflecting the underlying genetic liability to disease. In contrast, while age-at-menopause and age-at-menarche are highly polygenic, average effect sizes and the variance contributed by low-frequency variants are higher, supporting a history of stronger selection pressure in the human population[27].

Genome-wide association studies of time-to-event phenotypes are critical for gaining insights into the genetics of disease etiology and progression, but their application has been hampered by the computational and statistical challenges of the models, especially when the predictors are ultrahigh-dimensional. Our hybrid-parallel sampling scheme enables a fully Bayesian model to be applied to large-scale genomic data and we show improved genomic prediction over competing approaches, not only in the $R^2$ or C-statistic obtained but in the inference that can be obtained from a full posterior predictive

distribution. Previous evidence shows that cohort studies using proportional hazards (or Cox) regression models generally increase statistical power compared to case-control studies using logistic regression model[2,3]. Our results support this and we expect the benefits to become more evident as the number of cases accrues with accurate age-at-diagnosis information.

A typical approach in the time-to-event analysis is the Cox PH model[5] that uses a non-parametric estimate for the baseline hazard and then estimates other effect sizes proportional to this hazard. Our BayesW model is also a proportional hazard model with the constraint that the baseline hazard follows a Weibull distribution and thus marker effect size estimates have a similar interpretation as those from a Cox PH model. Interestingly, the results from both our simulation study and real data analysis show that when quantifying the significance of the markers and estimating the marker effect sizes, it might not be pivotal to capture the baseline hazard with a non-parametric method. The simulations show that even in the misspecified cases BayesW performs better compared to the semi-parametric Cox model, demonstrating that using a parametric assumption might be more descriptive than simply using a Cox PH model from standard practice.

There has been a significant amount of work on the heritability of the time-to-event traits. For example, it has been suggested to define heritability in the Weibull frailty model on the log-time scale[30], or on the log-frailty scale in Cox PH model[31]. Transforming the log-scale heritability to the original scale[32], has then required approximations and the term of original scale heritability has not been easy to explain and use[33]. Here, using a similar idea of partitioning the total phenotypic variance into genetic and error variance components, we present an expression for SNP heritability on the log-time scale. We then show that there exists a natural correspondence between log-scale and original scale heritability, without the need for any approximations, with log-scale and original scale heritability giving similar estimates if the Weibull shape parameter tends to higher values. Therefore, under Weibull assumptions, we provide a definition of SNP-heritability for time-to-event traits for the GWAS-era.

There are a number of key considerations and limitations. The assumption of a Weibull distribution for the traits considered here can induce bias in the hyperparameter estimates, although we have shown that this assumption yields accurate results in terms of prediction regardless of the phenotypic distribution. A third parameter could be introduced through the use of a generalised gamma distribution and this will be the focus of future work as it should allow for unbiased hyperparameter estimation irrespective of the trait distribution. In this study, we have used hard-coded genotypes to make the method computationally efficient which can result in reduced covariance between the imputed marker and the trait. However, we do not believe this to be a hindrance to our method or the application in this work as hard-coded genotypic values will likely be the norm with the upcoming release of whole-genome sequence data and our aim is to provide a time-to-event model that is capable of scaling to these data requirements. We apply our approach only to markers that are imputed in both the UK Biobank and the Estonian genome centre data and by selecting markers present in both populations we are favouring markers that impute well across human populations.

In addition, despite allowing for left-truncation in the likelihood, we focus on presenting a series of baseline results before extending our inference to account for differences in sampling, semi-competing risks across different outcomes, genomic annotation enrichment, and sex-differences both the effect sizes and in the sampling of different time-to-event outcomes all of which require extensions to the modelling framework, which are also the

focus of future work. Furthermore, we do not consider time-varying coefficients or time-varying covariates, which may improve inference as multiple measurements over time are collected in biobank studies. Nevertheless, this work represents the first step toward large-scale inference of the genomic basis of variation in the timing of common complex traits.

## Methods

**Parametrisation of Weibull distribution**. We define $Y_i$ as the time-to-event for individual $i$, with Weibull distribution $Y_i \sim W(a, b_i)$, where $a$ and $b_i$ are correspondingly the shape and scale parameters. The survival function is

$$S_i(y) = \exp\left(-\left(\frac{y}{b_i}\right)^a\right). \tag{3}$$

We are interested in modelling the mean and the variance of the time-to-event. Unfortunately, the mean and the variance of Weibull are dependent as they share both parameters in their expressions. Moreover, as the expressions for mean and variance contain gamma functions it is rather difficult to dissect the mean and variance to be dependant only on one parameter. One possible solution is to use $\log Y_i$ and its moments instead. If $Y_i \sim W(a, b_i)$ then $\log Y_i$ has a Gumbel distribution, with mean and variance

$$E(\log Y_i | b_i, a) = \log b_i - \frac{K}{a}, \tag{4}$$

$$\mathrm{Var}(\log Y_i | b_i, a) = \frac{\pi^2}{6a^2}. \tag{5}$$

where $K$ is the Euler-Mascheroni constant, $K \approx 0.57721$. The parametrisation for the variance is only dependent on one parameter which we denote as $\alpha = a$. As we are interested in modelling SNP effects $\boldsymbol{\beta}$, covariates $\boldsymbol{\delta}$ (sex, PCs) and the average scale for time-to-event $\mu$ (intercept), it is possible to introduce them in the following way $b_i = \exp(\mu + \mathbf{x}_i'\boldsymbol{\beta} + \mathbf{z}_i'\boldsymbol{\delta} + \frac{K}{\alpha})$, resulting in

$$E(\log Y_i | \mu, \boldsymbol{\beta}, \boldsymbol{\delta}, \alpha) = \mu + \mathbf{x}_i'\boldsymbol{\beta} + \mathbf{z}_i'\boldsymbol{\delta}, \tag{6}$$

$$\mathrm{Var}(\log Y_i | \mu, \boldsymbol{\beta}, \boldsymbol{\delta}, \alpha) = \frac{\pi^2}{6\alpha^2}, \tag{7}$$

where $\mathbf{x}_i$ is the vector of scaled SNP marker values and $\mathbf{z}_i$ is the vector of covariate values for an individual $i$ and $\pi = 3.14159\ldots$ is a constant. The third and the fourth moment for $\log Y_i$ are constant regardless of the parametrisation.

**Modelling time-to-event and age-at-onset**. As a baseline model, we propose to test the association of $Y_i$ with a series of covariates (SNP markers in this case) $X$ using a mixture of the regression model, with $\boldsymbol{\gamma}_j$ as the mixture indicator, with $\boldsymbol{\gamma}_j = k$ if $j$th marker is included in the $k$th mixture component of the model, $k \in \{1, \ldots, L\}$, and $\boldsymbol{\gamma}_j = 0$ if it is not included into the model. The expected value of time-to-event logarithm is then a linear combination of the markers included in the model plus the effect of the covariates and the intercept ($\mu$) as in Eq. (6) and error variance is expressed via the shape parameter as shown in Eq. (7). $\boldsymbol{\beta}_j$ have non-zero values if and only if $\boldsymbol{\gamma}_j \geq 1$. We assume that non-zero $\boldsymbol{\beta}_j$ from mixture component $k > 0$ ($\boldsymbol{\gamma}_j = k$) comes from a normal distribution with zero mean and variance $C_k \sigma_G^2$, that is $\boldsymbol{\beta}_j \sim N(0, C_k \sigma_G^2)$.

The survival and density function for $Y_i$ is correspondingly

$$S_i(y) = \exp\{-y^\alpha \exp(-\alpha(\mu + \mathbf{x}_i'\boldsymbol{\beta} + \mathbf{z}_i'\boldsymbol{\delta}) - K)\}, \tag{8}$$

$$f_i(y) = \exp\{-K - \alpha(\mu + \mathbf{x}_i'\boldsymbol{\beta} + \mathbf{z}_i'\boldsymbol{\delta}) - y^\alpha \exp(-\alpha(\mu + \mathbf{x}_i'\boldsymbol{\beta} + \mathbf{z}_i'\boldsymbol{\delta}) - K)\}y^{\alpha-1}\alpha, \tag{9}$$

The likelihood function for the right-censored and left truncated data of $n$ individuals is then

$$
\begin{aligned}
p(D|\alpha, \boldsymbol{\beta}, \boldsymbol{\delta}, \mu) &= \prod_{i=1}^{n} \frac{1}{S(a_i)} \prod_{i=1}^{n} f(y_i)^{d_i} S(y_i)^{1-d_i} \\
&= \alpha^d \exp\Bigg\{ -Kd + (\alpha - 1)\sum_{i=1}^{n} d_i \log y_i - \alpha \sum_{i=1}^{n} d_i(\mu + \mathbf{x}_i'\boldsymbol{\beta} + \mathbf{z}_i'\boldsymbol{\delta}) - \\
&\quad - \sum_{i=1}^{n} y_i^\alpha \exp(-\alpha(\mu + \mathbf{x}_i'\boldsymbol{\beta} + \mathbf{z}_i'\boldsymbol{\delta}) - K) + \sum_{i=1}^{n} a_i^\alpha \exp(-\alpha(\mu + \mathbf{x}_i'\boldsymbol{\beta} + \mathbf{z}_i'\boldsymbol{\delta}) - K) \Bigg\},
\end{aligned}
\tag{10}
$$

where $d_i$ is the failure indicator and $d$ is the number of events at the end of the periods; $a_i$ is the time of left truncation. It is possible to use the model without left truncation. In order to do so, for every $i$, we will assume that $a_i = 0$. Whenever $a_i = 0$, we will naturally define $\exp(\alpha(\log(a_i) - \mu - \mathbf{x}_i'\boldsymbol{\beta} - \mathbf{z}_i'\boldsymbol{\delta})) = 0$, thus the left truncation would not contribute to the likelihood in the Eq. (10).

Let the prior distribution of $\alpha$ be a gamma distribution with parameters $\alpha_0, \kappa_0$

$$p(\alpha) \propto \alpha^{\alpha_0 - 1} \exp(-\kappa_0 \alpha), \tag{11}$$

the prior for $\boldsymbol{\beta}_j$ be normal:

$$p(\boldsymbol{\beta}_j|\sigma_G^2,\boldsymbol{\gamma}_j=k)\propto\left(\frac{1}{C_k\sigma_G^2}\right)^{0.5}\exp\left[-\frac{1}{2C_k\sigma_G^2}\boldsymbol{\beta}_j^2\right], \qquad (12)$$

the prior for $\sigma_G^2$ being inverse gamma distribution with parameters $\alpha_\sigma$ and $\beta_\sigma$

$$p(\sigma_G^2)\propto\left(\frac{1}{\sigma_G^2}\right)^{\alpha_\sigma+1}\exp\left(-\frac{\beta_\sigma}{\sigma_G^2}\right), \qquad (13)$$

the prior for $\boldsymbol{\delta}_q$ ($q$th covariate) be normal with variance parameter $\sigma_\delta^2$:

$$p(\boldsymbol{\delta}_q)\propto\exp\left(-\frac{1}{2\sigma_\delta^2}\boldsymbol{\delta}_q^2\right), \qquad (14)$$

the prior for $\mu$ be normal with variance parameter $\sigma_\mu^2$:

$$p(\mu)\propto\exp\left(-\frac{1}{2\sigma_\mu^2}\mu^2\right), \qquad (15)$$

the prior for $\boldsymbol{\gamma}_j$ be multinomial:

$$p(\boldsymbol{\gamma}_j|\boldsymbol{\pi})=\pi_0^{I(\gamma_j=0)}\cdot\,...\,\cdot\pi_L^{I(\gamma_j=L)}, \qquad (16)$$

the prior probabilities of belonging to each of the mixture distributions $k$ are stored in $L+1$-dimensional vector $\boldsymbol{\pi}$ with the prior for $\boldsymbol{\pi}$ a Dirichlet distribution

$$p(\boldsymbol{\pi})=\text{Dirichlet}\,(\mathbf{p}_L), \qquad (17)$$

where $I(\cdot)$ is the indicator function and $\mathbf{p}_L$ is the $L+1$-dimensional vector with prior values. For the exact values of the prior specification see Data Analysis Details.

The conditional posterior distribution for $\sigma_G^2$ is inverse gamma with parameters

$$\alpha_\sigma+0.5\sum_{k=1}^{L}|\gamma^k|\;\text{ and }\;0.5\sum_{k=1}^{L}|\gamma^k|\boldsymbol{\beta}'_{\gamma^k}\boldsymbol{\beta}_{\gamma^k}+\beta_\sigma,$$

where $\gamma^k$ denotes the set of indices $j$ for which $\gamma_j=k$. The conditional posterior distribution for $\boldsymbol{\pi}$ is Dirichlet distribution

$$p(\boldsymbol{\pi}|\boldsymbol{\gamma})=\text{Dirichlet}\,(\mathbf{p}_L+(|\gamma^0|,...,|\gamma^L|)). \qquad (18)$$

Unfortunately, there is no simple form for the conditional posteriors of $\alpha,\mu,\boldsymbol{\beta}_j$ and $\boldsymbol{\delta}_q$. However, the conditional posterior distributions are log-concave (see Supplementary Note), and thus the sampling for $\alpha,\mu,\boldsymbol{\beta}_j$ and $\boldsymbol{\delta}_q$ can be conducted using adaptive rejection sampling requiring only the log posteriors. Denoting $\boldsymbol{\beta}_{-j}$ as all the $\boldsymbol{\beta}$ parameters, excluding $\boldsymbol{\beta}_j$, and $\boldsymbol{\delta}_{-q}$ as all the $\boldsymbol{\delta}$ parameters, excluding $\boldsymbol{\delta}_q$, these are

$$\log p(\alpha|D,\mu,\boldsymbol{\beta},\boldsymbol{\delta})=\text{const}+(\alpha_0+d-1)\log\alpha+\alpha\left[\sum_{i=1}^{n}d_i(\log y_i-\mu-\mathbf{x}'_i\boldsymbol{\beta}-\mathbf{z}'_i\boldsymbol{\delta})-\kappa_0\right]$$
$$+\exp(-K)\sum_{i=1}^{n}[\exp(\alpha(\log(a_i)-\mu-\mathbf{x}'_i\boldsymbol{\beta}-\mathbf{z}'_i\boldsymbol{\delta}))-\exp(\alpha(\log(y_i)-\mu-\mathbf{x}'_i\boldsymbol{\beta}-\mathbf{z}'_i\boldsymbol{\delta}))], \qquad (19)$$

$$\log p(\boldsymbol{\beta}_j|D,\alpha,\mu,\boldsymbol{\beta}_{-j},\boldsymbol{\delta},\sigma_G^2,\boldsymbol{\gamma}_j=k)=\log\left(\frac{\text{const}}{\sqrt{C_k}}\right)-\alpha\boldsymbol{\beta}_j\sum_{i=1}^{n}d_i\mathbf{x}_{ij}$$
$$+\exp(-K)\sum_{i=1}^{n}[\exp(\alpha(\log(a_i)-\mu-\mathbf{x}'_i\boldsymbol{\beta}-\mathbf{z}'_i\boldsymbol{\delta}))-\exp(\alpha(\log(y_i)-\mu-\mathbf{x}'_i\boldsymbol{\beta}-\mathbf{z}'_i\boldsymbol{\delta}))]-\frac{1}{2C_k\sigma_G^2}\boldsymbol{\beta}_j^2, \qquad (20)$$

$$\log p(\boldsymbol{\delta}_q|D,\alpha,\mu,\boldsymbol{\beta},\boldsymbol{\delta}_{-q},\sigma_\delta^2)=-\alpha\boldsymbol{\delta}_q\sum_{i=1}^{n}d_i\mathbf{z}_{ij}$$
$$+\exp(-K)\sum_{i=1}^{n}[\exp(\alpha(\log(a_i)-\mu-\mathbf{x}'_i\boldsymbol{\beta}-\mathbf{z}'_i\boldsymbol{\delta}))-\exp(\alpha(\log(y_i)-\mu-\mathbf{x}'_i\boldsymbol{\beta}-\mathbf{z}'_i\boldsymbol{\delta}))]-\frac{1}{2\sigma_\delta^2}\boldsymbol{\delta}_q^2, \qquad (21)$$

$$\log p(\mu|D,\alpha,\boldsymbol{\beta},\boldsymbol{\delta})=\text{const}-\alpha\mu d$$
$$+\exp(-K)\sum_{i=1}^{n}[\exp(\alpha(\log(a_i)-\mu-\mathbf{x}'_i\boldsymbol{\beta}-\mathbf{z}'_i\boldsymbol{\delta}))-\exp(\alpha(\log(y_i)-\mu-\mathbf{x}'_i\boldsymbol{\beta}-\mathbf{z}'_i\boldsymbol{\delta}))]-\frac{1}{2\sigma_\mu^2}\mu^2. \qquad (22)$$

**Selection of the mixture component**. We intend to do variable selection and select mixture components by using the idea of spike and slab priors[34], where the spike part of the prior has a point mass of 0. SNP will be assigned to a mixture component by comparing the ratios of the marginal likelihood. For mixture selection for the $j$th SNP, we need to find the following marginal likelihood for every $k$. Suppose here that $C_0>0$ is the factor for the 0th mixture (spike)

$$p(D|\boldsymbol{\beta}_{-j},\boldsymbol{\delta},\sigma_G^2,\alpha,\mu,\boldsymbol{\gamma}_j=k)=\int_{\boldsymbol{\beta}_j}p(D|\boldsymbol{\beta}_{-j},\boldsymbol{\delta},\mu,\alpha,\boldsymbol{\beta}_j)p(\boldsymbol{\beta}_j|\sigma_G^2,\boldsymbol{\gamma}_j=k)d\boldsymbol{\beta}_j$$
$$=\frac{Q}{\sqrt{C_k}}\int_{\boldsymbol{\beta}_j}\exp\{h_k(\boldsymbol{\beta}_j)\}d\boldsymbol{\beta}_j, \qquad (23)$$

where $D$ represents the observed data, $Q$ is a positive constant that is not

dependent on $k$ and

$$h_k(\boldsymbol{\beta}_j)=-\alpha\boldsymbol{\beta}_j\sum_{i=1}^{n}d_i\mathbf{x}_{ij}$$
$$+\exp(-K)\sum_{i=1}^{n}[\exp(\alpha(\log(a_i)-\mu-\mathbf{x}'_i\boldsymbol{\beta}-\mathbf{z}'_i\boldsymbol{\delta}))-\exp(\alpha(\log(y_i)-\mu-\mathbf{x}'_i\boldsymbol{\beta}-\mathbf{z}'_i\boldsymbol{\delta}))]-\frac{1}{2C_k\sigma_G^2}\boldsymbol{\beta}_j^2. \qquad (24)$$

The probability for $\boldsymbol{\gamma}_j$ is :

$$p(\boldsymbol{\gamma}_j=k|D,\boldsymbol{\beta}_{-j},\boldsymbol{\delta},\sigma_G^2,\alpha,\mu)=Cp(D|\boldsymbol{\beta}_{-j},\boldsymbol{\delta},\boldsymbol{\gamma}_j=k,\sigma_G^2,\alpha,\mu)p(\boldsymbol{\gamma}_j=k). \qquad (25)$$

where $C$ is a positive constant that is not dependent on $k$. Denoting $\Theta=\{D,\boldsymbol{\beta}_{-j},\boldsymbol{\delta},\sigma_G^2,\alpha,\mu\}$, the probability to include SNP $j$ in the component $k$ can be calculated as

$$p(\boldsymbol{\gamma}_j=k|\Theta)=\frac{Cp(\boldsymbol{\gamma}_j=k|\Theta)}{C(p(\boldsymbol{\gamma}_j=0|\Theta)+...+p(\boldsymbol{\gamma}_j=L|\Theta))}. \qquad (26)$$

For every $k$

$$p(\boldsymbol{\gamma}_j=k|\Theta)=\frac{\frac{\pi_k}{\sqrt{C_k}}\int_{\boldsymbol{\beta}_j}\exp(h_k(\boldsymbol{\beta}_j))d\boldsymbol{\beta}_j}{\frac{\pi_0}{\sqrt{C_0}}\int_{\boldsymbol{\beta}_j}\exp(h_0(\boldsymbol{\beta}_j))d\boldsymbol{\beta}_j+...+\frac{\pi_L}{\sqrt{C_L}}\int_{\boldsymbol{\beta}_j}\exp(h_L(\boldsymbol{\beta}_j))d\boldsymbol{\beta}_j} \qquad (27)$$

Here, the numerator represents the marginal likelihood assuming $j$th variable is included in the $k$th mixture component.

In general, it is not possible to find an analytic expression for the integrals presented in Eq. (23), thus some numeric method has to be used for approximating their values. For this, we use adaptive Gauss–Hermite quadrature as the integral is improper with infinite endpoints.

We start by rewriting the expression (Eq. 24) as

$$h_k(\boldsymbol{\beta}_j)=-\alpha\boldsymbol{\beta}_j\sum_{i=1}^{n}d_i\mathbf{x}_{ij}+\sum_{i=1}^{n}[\exp(u_i-\alpha\mathbf{x}_{ij}\boldsymbol{\beta}_j)-\exp(v_i-\alpha\mathbf{x}_{ij}\boldsymbol{\beta}_j)]-\frac{1}{2C_k\sigma_G^2}\boldsymbol{\beta}_j^2, \qquad (28)$$

where $v_i=\alpha(\log y_i-\mu-\mathbf{x}'_{-j}\boldsymbol{\beta}_{-j}-\mathbf{z}'_i\boldsymbol{\delta})-K$ and $u_i$ is analogous with $a_i$ instead of $y_i$. We introduce a reparameterisation with variable $s$

$$s=\frac{\boldsymbol{\beta}_j}{\sqrt{2C_k\sigma_G^2}} \qquad (29)$$

and therefore we get from Eq. (23)

$$Q\int\frac{1}{\sqrt{C_k}}\exp\{h_k(\boldsymbol{\beta}_j)\}d\boldsymbol{\beta}_j$$
$$=Q\int\frac{\sqrt{2C_k\sigma_G^2}}{\sqrt{C_k}}\exp\left\{-\alpha s\sqrt{2C_k\sigma_G^2}\sum_{i=1}^{n}d_i\mathbf{x}_{ij}+\sum_{i=1}^{n}\left[\exp(u_i-\alpha\mathbf{x}_{ij}s\sqrt{2C_k\sigma_G^2})-\exp(v_i-\alpha\mathbf{x}_{ij}s\sqrt{2C_k\sigma_G^2})\right]-s^2\right\}ds$$
$$=Q\sqrt{2\sigma_G^2}\int\exp\left\{-\alpha s\sqrt{2C_k\sigma_G^2}\sum_{i=1}^{n}d_i\mathbf{x}_{ij}+\sum_{i=1}^{n}\left[\exp(u_i-\alpha\mathbf{x}_{ij}s\sqrt{2C_k\sigma_G^2})-\exp(v_i-\alpha\mathbf{x}_{ij}s\sqrt{2C_k\sigma_G^2})\right]-s^2\right\}ds$$
$$=Q\sqrt{2\sigma_G^2}\int g_k(s)ds. \qquad (30)$$

in the last expression in Eq. (27), the term $Q\sqrt{2\sigma_G^2}$ cancels out from the numerator and the denominator.

If the smallest mixture variance factor $C_0>0$, then the corresponding spike distribution is absolutely continuous. As we would like to use Dirac spike instead, we define the corresponding marginal likelihood by finding the limit of the expression in the process $C_0\to 0+$.

$$p(D|\boldsymbol{\beta}_{-j},\boldsymbol{\delta},\sigma_G^2,\alpha,\mu,\boldsymbol{\gamma}_j=0)=\lim_{C_0\to 0+}Q\int\frac{1}{\sqrt{C_0}}\exp\{h_0(\boldsymbol{\beta}_j)\}d\boldsymbol{\beta}_j. \qquad (31)$$

We are only interested in $C_0$ in the limiting process so without the loss of generality we define $C_0$ through an auxiliary positive integer variable $l$ as $C_0=\frac{1}{l}$ and using the reparametrisation result from Eq. (30) we get that

$$p(D|\boldsymbol{\beta}_{-j},\boldsymbol{\delta},\sigma_G^2,\alpha,\mu,\boldsymbol{\gamma}_j=0)$$
$$=\lim_{C_0\to 0+}Q\sqrt{2\sigma_G^2}\int\exp\left\{-\alpha s\sqrt{2C_0\sigma_G^2}\sum_{i=1}^{n}d_i\mathbf{x}_{ij}+\sum_{i=1}^{n}\left[\exp(u_i-\alpha\mathbf{x}_{ij}s\sqrt{2C_0\sigma_G^2})-\exp(v_i-\alpha\mathbf{x}_{ij}s\sqrt{2C_0\sigma_G^2})\right]-s^2\right\}ds$$
$$=\lim_{l\to\infty}Q\sqrt{2\sigma_G^2}\int\exp\left\{-\alpha s\sqrt{2\sigma_G^2/l}\sum_{i=1}^{n}d_i\mathbf{x}_{ij}+\sum_{i=1}^{n}\left[\exp(u_i-\alpha\mathbf{x}_{ij}s\sqrt{2\sigma_G^2/l})-\exp(v_i-\alpha\mathbf{x}_{ij}s\sqrt{2\sigma_G^2/l})\right]-s^2\right\}ds$$
$$=\lim_{l\to\infty}Q\sqrt{2\sigma_G^2}\int f(s,l)\exp\{-s^2\}ds. \qquad (32)$$

As $f(s,l)\leq 1$ for every possible combination of arguments, because in the data censoring or event occurs only after entering the study, we can write that

$$f(s,l)\exp\{-s^2\}\leq\exp\{-s^2\},\forall l \qquad (33)$$

which means that the integrand in Eq. (32) is dominated by $\exp\{-s^2\}$. Furthermore, we see that the limit of the integrand is

$$\lim_{l\to\infty}f(s,l)\exp\{-s^2\}=\exp\left\{\sum_{i=1}^{n}[\exp(u_i)-\exp(v_i)]-s^2\right\}. \qquad (34)$$

As $\int\exp\{-s^2\}ds<\infty$, it is possible to use the Lebesgue's dominated convergence

theorem and therefore

$$\lim_{l \to \infty} Q\sqrt{2\sigma_G^2} \int f(s, l) \exp\{-s^2\} ds = Q\sqrt{2\sigma_G^2} \int \exp\left\{\sum_{i=1}^{n}[\exp(u_i) - \exp(v_i)] - s^2\right\} ds$$

$$= Q\sqrt{2\sigma_G^2} \exp\left\{\sum_{i=1}^{n}[\exp(u_i) - \exp(v_i)]\right\} \int \exp\{-s^2\} ds = Q\sqrt{2\pi\sigma_G^2} \exp\left\{\sum_{i=1}^{n}[\exp(u_i) - \exp(v_i)]\right\}.$$

(35)

In conclusion, we have derived the expression for the marginal likelihood for the Dirac spike variance component as

$$p(D|\boldsymbol{\beta}_{-j}, \boldsymbol{\delta}, \sigma_G^2, \alpha, \mu, \boldsymbol{\gamma}_j = 0) = Q\sqrt{2\pi\sigma_G^2} \exp\left\{\sum_{i=1}^{n}[\exp(u_i) - \exp(v_i)]\right\}. \quad (36)$$

**Adaptive Gauss-Hermite quadrature.** It is possible to use Gauss-Hermite quadrature, however, it can happen that for adequate precision one has to use a large number of quadrature points leading to more calculations. Adaptive Gauss-Hermite quadrature can make the procedure more efficient. For any function $g_k(s)$ as defined in Eq. (30), we can write

$$\int_{-\infty}^{\infty} g_k(s) ds \approx \hat{\sigma}\sqrt{2} \sum_{r=1}^{m} w_r g_k(\hat{\mu} + \hat{\sigma}\sqrt{2}t_r), \quad (37)$$

where $\hat{\mu}$ could be chosen as the mode of $g_k(s)$ and $\hat{\sigma} = \frac{1}{\sqrt{-(\log g_k(s))''|_{s=\hat{\mu}}}}$; $m$ is the number of quadrature points, $t_r$ is the roots of $m$th order Hermite polynomial and $w_r$ are corresponding weights[35].

Finding the posterior mode can be computationally cumbersome, calculating $\hat{\sigma}$ requires evaluating the logarithm of $g_k$ at this mode. As we assume that the effects sizes are distributed symmetrically around zero, we use $\hat{\mu} = 0$ which avoids numerical posterior mode calculations and evaluating the second derivative at different posterior modes.

**Posterior inclusion probability.** Combining the previous results we get a numerical solution for calculating the posterior inclusion probability. For every $k > 0$ the inclusion probabilities are

$$p(\boldsymbol{\gamma}_j = k|\Theta) = \frac{\pi_k \int g_k(s) ds}{\pi_0 \sqrt{\pi} \exp\{\sum_{i=1}^{n}[\exp(u_i) - \exp(v_i)]\} + \sum_{l=1}^{L} \pi_l \int g_l(s) ds}$$

$$\approx \frac{\pi_k \hat{\sigma}_k \sqrt{2} \sum_{r=1}^{m} w_r g_k(\hat{\sigma}_k \sqrt{2}t_r)}{\pi_0 \sqrt{\pi}T + \sum_{l=1}^{L} \pi_l \hat{\sigma}_l \sqrt{2} \sum_{r=1}^{m} w_r g_l(\hat{\sigma}_l \sqrt{2}t_r)} = \frac{\pi_k \sqrt{2}\hat{\sigma}_k \sum_{r=1}^{m} w_r g_k(\hat{\sigma}_k \sqrt{2}t_r)/T}{\pi_0 \sqrt{\pi} + \sum_{l=1}^{L} \pi_l \hat{\sigma}_l \sqrt{2} \sum_{r=1}^{m} w_r g_l(\hat{\sigma}_l \sqrt{2}t_r)/T}.$$

(38)

Similarly, we can find the probability of excluding ($\boldsymbol{\gamma}_j = 0$) the marker

$$p(\boldsymbol{\gamma}_j = 0|\Theta) = \frac{\pi_0 \sqrt{\pi}}{\pi_0 \sqrt{\pi} + \sum_{l=1}^{L} \pi_l \hat{\sigma}_l \sqrt{2} \sum_{r=1}^{m} w_r g_l(\hat{\sigma}_l \sqrt{2}t_r)/T}. \quad (39)$$

Both cases $\hat{\sigma}_k$ are calculated as

$$\hat{\sigma}_k = \frac{1}{\sqrt{-(\log g_k(s))''|_{s=0}}} = \frac{1}{\sqrt{2}\sqrt{1 + \alpha^2 C_k \sigma_G^2 \sum_{i=1}^{n} \mathbf{x}_{ij}^2 (\exp(v_i) - \exp(u_i))}}. \quad (40)$$

For computational purposes, we evaluate $\frac{g_k(\hat{\sigma}_k \sqrt{2}t_r)}{T}$ as

$$\frac{g_k(\hat{\sigma}_k \sqrt{2}t_r)}{T} = \exp\left\{-\alpha\hat{\sigma}_k \sqrt{2}t_r \sqrt{2C_k \sigma_G^2} \sum_{i=1}^{n} d_i \mathbf{x}_{ij}\right.$$
$$+ \sum_{i=1}^{n} \left[\exp(u_i - \alpha\mathbf{x}_{ij}\hat{\sigma}_k \sqrt{2}t_r \sqrt{2C_k \sigma_G^2}) - \exp(u_i) + \exp(v_i) - \exp(v_i - \alpha\mathbf{x}_{ij}\hat{\sigma}_k \sqrt{2}t_r \sqrt{2C_k \sigma_G^2})\right] - (\hat{\sigma}_k \sqrt{2}t_r)^2\right\}$$
$$= \exp\left\{-\alpha\hat{\sigma}_k \sqrt{2}t_r \sqrt{2C_k \sigma_G^2} \sum_{i=1}^{n} d_i \mathbf{x}_{ij} + \sum_{i=1}^{n} \left[(\exp(v_i) - \exp(u_i))(1 - \exp(-\alpha\mathbf{x}_{ij}\hat{\sigma}_k \sqrt{2}t_r \sqrt{2C_k \sigma_G^2}))\right] - (\hat{\sigma}_k \sqrt{2}t_r)^2\right\}. \quad (41)$$

**Adaptive rejection sampling.** To sample $\alpha$, $\mu$ and $\boldsymbol{\beta}_j$, $\boldsymbol{\delta}_\varphi$, we use Adaptive Rejection Sampling, initially outlined by Gilks and Wild[36]. The prerequisite of the method is the log-concavity of the sampled density function.

The idea of the method is to build an envelope around the log-density. The lower hull is constructed by evaluating the function at some pre-specified abscissae and connecting the evaluation results with linear functions resulting in a piece-wise linear lower hull. The upper hull can be constructed either by using tangents evaluated at the prespecified abscissae (Derivative based ARS) or by extending the linear functions obtained in the construction of the lower hull (Derivative free ARS[37]). Although the derivative-based method might result in a more accurate upper hull, thus leading to faster sampling, it would still require evaluating derivatives and thus we employ the derivative-free method.

The proposals are sampled from an appropriately scaled exponent of the upper hull from which it is easier to sample. The sampling proposal will go through tests. If the proposal is not accepted then it will be included in the set of used abscissae to create a more accurate envelope in the next round. Therefore, the method requires specifying the log posterior and at least three initial abscissae. It also requires some abscissae larger and smaller than the posterior mode. To set the abscissae for some parameter $\theta$, we could, for example, choose the abscissae $\{\hat{\theta} - c_\theta, \hat{\theta}, \hat{\theta} + c_\theta\}$, where $\hat{\theta}$ is ideally the posterior mode for $\theta$. $c_\theta$ is some positive number that would guarantee that at least one of the proposed abscissae would be larger than posterior

mode and one smaller. If $\hat{\theta}$ is the posterior mode, then $c_\theta$ choice is arbitrary and a smaller $c_\theta$ is preferred, potentially decreasing the sampling time.

In addition, the derivative-free method requires specifying the minimum and maximum value of the distribution, an assumption that is often incorrect. In practice, it poses no problems as we can simply set the required minima and maxima to be extreme enough so that the distribution is very unlikely to reach those values. To sample intercept $\mu$ we set the limits to 2 and 5 which after exponentiation would correspond to 7.39 and 148.4 which we believe each of our posterior means should fit in; for $\alpha$ we set the limits to 0 to 40; for non-zero betas, we used the previous beta value $\pm 2\sqrt{C_k \sigma_G^2}$ as minimum and maximum limits for sampling as this can adapt to different mixtures and should still safely retain almost the entire posterior distribution. The Adaptive Rejection Sampling was implemented in C code by Gilks (http://www1.maths.leeds.ac.uk/~wally.gilks/adaptive.rejection/web_page/Welcome.html, accessed 26.08.2020). In the Supplementary Note, we provide proof of the log-concavity of the functions sampled.

**Sampling algorithm.** We summarise the serial sampling algorithm in Algorithm 29 along with the specification for the prior distributions and the initialisation of the model parameters. Algorithm 2 summarises the Bulk Synchronous Gibbs sampling for BayesW that extends Algorithm 29. If the number of workers $T = 1$ and the synchronisation rate $u = 1$ then Algorithm 2 reduces down to Algorithm 29.

**Extension to a grouped mixture of regressions model.** Here, we now assume that the SNP marker effects come from $\Phi$ of disjoint groups, with a reparametrisation of the model parameters to represent the mean of the logarithm of the phenotype as

$$E(\log Y_i | \mu, \boldsymbol{\beta}, \boldsymbol{\delta}, \alpha) = \mu + \sum_{\varphi=1}^{\Phi}(\mathbf{x}_i^\varphi)' \boldsymbol{\beta}^\varphi + \mathbf{z}_i' \boldsymbol{\delta}, \quad (42)$$

where there is a single intercept term $\mu$, and a single Weibull shape parameter $\alpha$, but now $\mathbf{x}_i^\varphi$ are the standardised marker values in group $\varphi$, $\boldsymbol{\beta}^\varphi$ are the marker estimates for the corresponding group. Each $\boldsymbol{\beta}_j^\varphi$ is distributed according to:

$$\beta_j^\varphi \sim \pi_0^\varphi \boldsymbol{\delta}_0 + \pi_1^\varphi \mathcal{N}(0, C_1^\varphi \sigma_{G\varphi}^2) + \pi_2^\varphi \mathcal{N}(0, C_2^\varphi \sigma_{G\varphi}^2) + \ldots + \pi_{L_\varphi}^\varphi \mathcal{N}(0, C_L^\varphi \sigma_{G\varphi}^2) \quad (43)$$

where for each SNP marker group prior probabilities of belonging to each of the mixture distribution $k$ is stored in $L_\varphi + 1$-dimensional vector $\boldsymbol{\pi}^\varphi$ and these mixture proportions $\{\pi_0^\varphi, \pi_1^\varphi, \ldots, \pi_L^\varphi\}$, $\sum_{k=0}^{L} \pi_k^\varphi = 1$ are updated in each iteration. Each mixture component ($\boldsymbol{\gamma}_j = k \geq 1$) is a normal distribution with zero mean and variance $C_k^\varphi \sigma_{G\varphi}^2$, where $\sigma_{G\varphi}^2$ represents the phenotypic variance attributable to markers of group $\varphi$ and $C_k^\varphi$ is group and mixture specific factor showing the magnitude of variance explained by this specific mixture. Thus, the mixture proportions, variance explained by the SNP markers, and mixture constants are all unique and independent across SNP marker groups. The formulation presented here of having an independent variance parameter $\sigma_{G\varphi}^2$ per group of markers, and independent mixture variance components enable estimation of the amount of phenotypic variance attributable to the group-specific effects and enable differences in the distribution of effects among groups. All of the steps shown in previous paragraphs are still valid and now we are using group-specific genetic variances $\sigma_{G\varphi}^2$, prior inclusion probabilities $\boldsymbol{\pi}^\varphi$ and mixture proportions $C_k^\varphi$. Furthermore, due to the fact that the model is additive, the sum of group-specific genetic variances represents the total genetic variance $\sigma_G^2 = \sum_{\varphi=1}^{\Phi} \sigma_{G\varphi}^2$.

**Derivations for the sparse calculations.** In order to reduce the number of computations and improve running times we derive a sparse representation of genotypes, given that conditional posterior distributions in our scheme are different, we have to derive different update equations. Suppose $\xi_{ij}$ represents the $j$th SNP allele count (0, 1 or 2) for the $i$th individual, and $\bar{\xi}_j$, $s_j$ represent the mean and standard deviation of the $j-$th SNP in our sample. In the regular setting we would like to use standardised count values ($\mathbf{x}_{ij} = \frac{\xi_{ij} - \bar{\xi}_j}{s_j}$) instead and meanwhile, speed up the computations by using the knowledge that $\mathbf{x}_{ij}$ can have only three values within an SNP.

There are three equations where we can apply sparsity. Firstly, Eq. (40) for the $\hat{\sigma}_k$ term (for the $j$th SNP) in the adaptive Gauss-Hermite quadrature can be

expressed as

$$
\hat{\sigma}_k = \frac{1}{\sqrt{2}} \left[ 1 + \alpha C_k \sigma_G^2 \frac{1}{s_j^2} \sum_{i=1}^{n} \left( \frac{\xi_{ij} - \bar{\xi}_j}{s_j} \right)^2 (\exp(v_i) - \exp(u_i)) \right]^{-0.5}
$$

$$
= \frac{1}{\sqrt{2}} \left[ 1 + \frac{\alpha C_k \sigma_G^2}{s_j^2} \left( \sum_{i=1}^{n} \xi_{ij}^2 (\exp(v_i) - \exp(u_i)) - 2\bar{\xi}_j \sum_{i=1}^{n} \xi_{ij} (\exp(v_i) - \exp(u_i)) + \bar{\xi}_j^2 \sum_{i=1}^{n} (\exp(v_i) - \exp(u_i)) \right) \right]^{-0.5}
$$

$$
= \frac{1}{\sqrt{2}} \left[ 1 + \frac{\alpha C_k \sigma_G^2}{s_j^2} \left( \sum_{\xi_{ij}=1} (\exp(v_i) - \exp(u_i)) + 4 \sum_{\xi_{ij}=2} (\exp(v_i) - \exp(u_i)) \right. \right.
$$

$$
\left. \left. - 2\bar{\xi}_j \sum_{\xi_{ij}=1} (\exp(v_i) - \exp(u_i)) - 4\bar{\xi}_j \sum_{\xi_{ij}=2} (\exp(v_i) - \exp(u_i)) + \bar{\xi}_j^2 \sum_{i=1}^{n} (\exp(v_i) - \exp(u_i)) \right) \right]^{-0.5}
$$

$$
= \frac{1}{\sqrt{2}} \left[ 1 + \frac{\alpha C_k \sigma_G^2}{s_j^2} \left( (1 - 2\bar{\xi}_j) \sum_{\xi_{ij}=1} (\exp(v_i) - \exp(u_i)) + 4(1 - \bar{\xi}_j) \sum_{\xi_{ij}=2} (\exp(v_i) - \exp(u_i)) + \bar{\xi}_j^2 \sum_{i=1}^{n} (\exp(v_i) - \exp(u_i)) \right) \right]^{-0.5}.
$$

(44)

We see that $s_j$ and $\bar{\xi}_j$ and the expressions containing these terms can be calculated already beforehand for each SNP $j$.

Secondly, we can use the knowledge about sparsity to simplify expression (Eq. 41). More specifically

$$
\sum_{i=1}^{n} [(\exp(v_i) - \exp(u_i))(1 - \exp(-\alpha \mathbf{x}_{ij} \hat{\sigma}_k \sqrt{2} t_r \sqrt{2 C_k \sigma_G^2}))]
$$

$$
= \sum_{i=1}^{n} [(\exp(v_i) - \exp(u_i))] - \exp\left( \frac{\alpha \bar{\xi}_j}{s_j} \hat{\sigma}_k \sqrt{2} t_r \sqrt{2 C_k \sigma_G^2} \right) \sum_{i=1}^{n} [(\exp(v_i) - \exp(u_i))] \exp\left( -\frac{\alpha \xi_{ij}}{s_j} \hat{\sigma}_k \sqrt{2} t_r \sqrt{2 C_k \sigma_G^2} \right)
$$

$$
= \sum_{i=1}^{n} [(\exp(v_i) - \exp(u_i))] - \exp\left( \frac{\alpha \bar{\xi}_j}{s_j} \hat{\sigma}_k \sqrt{2} t_r \sqrt{2 C_k \sigma_G^2} \right) \cdot \left[ \sum_{\xi_{ij}=0} (\exp(v_i) - \exp(u_i)) \right.
$$

$$
\left. + \exp\left( -\frac{\alpha}{s_j} \hat{\sigma}_k \sqrt{2} t_r \sqrt{2 C_k \sigma_G^2} \right) \sum_{\xi_{ij}=1} (\exp(v_i) - \exp(u_i)) + \exp\left( -\frac{2\alpha}{s_j} \hat{\sigma}_k \sqrt{2} t_r \sqrt{2 C_k \sigma_G^2} \right) \sum_{\xi_{ij}=2} (\exp(v_i) - \exp(u_i)) \right].
$$

(45)

Thirdly, in expression (Eq. 20) we can rewrite the transformed residuals as

$$
\sum_{i=1}^{n} [\exp(\alpha(\log(a_i) - \mu - \mathbf{x}_i'\boldsymbol{\beta} - \mathbf{z}_i'\boldsymbol{\delta})) - \exp(\alpha(\log(y_i) - \mu - \mathbf{x}_i'\boldsymbol{\beta} - \mathbf{z}_i'\boldsymbol{\delta}))]
$$

$$
= \sum_{i=1}^{n} \left[ \exp\left( u_i - \alpha \left( \frac{\xi_{ij} - \bar{\xi}_j}{s_j} \right) \beta_j \right) - \exp\left( v_i - \alpha \left( \frac{\xi_{ij} - \bar{\xi}_j}{s_j} \right) \beta_j \right) \right] = \exp\left( \frac{\alpha \beta_j \bar{\xi}_j}{s_j} \right) \cdot
$$

$$
\left[ - \sum_{\xi_{ij}=0} (\exp(v_i) - \exp(u_i)) - \exp\left( -\frac{\alpha \beta_j}{s_j} \right) \sum_{\xi_{ij}=1} (\exp(v_i) - \exp(u_i)) - \exp\left( -\frac{2\alpha \beta_j}{s_j} \right) \sum_{\xi_{ij}=2} (\exp(v_i) - \exp(u_i)) \right].
$$

(46)

For all cases, after each update, we need to recalculate the difference $\exp(v_i) - \exp(u_i)$ for each individual $i$. We notice that it is sufficient to use three sums ($\sum_{\xi_{ij}=\xi} (\exp(v_i) - \exp(u_i))$, $\xi \in \{0, 1, 2\}$) that we denote as $V_0^j$, $V_1^j$, $V_2^j$ which are used in both of the final expressions. Thus, we have eliminated the need to calculate exponents and dot products in expressions (Eq. 40) and (Eq. 41), reducing them to a series of sparse summations and making the analysis scale sub-linearly with increasing marker number.

**Simulation study.** We conducted simulations to analyse the performance of our model under model misspecification, where the phenotypic distribution does not conform to a Weibull distribution, and to different censoring levels in the data. We assessed (i) estimation of hyperparameters, (ii) false discovery rate, and (iii) prediction accuracy. We used $M = 50,000$ uncorrelated markers and $N = 5000$ individuals for whom we simulated effects on $p = 500$ randomly selected markers, heritability (as defined in the Supplementary Note) was set to be $h^2 = 0.5$. Then, we generated phenotypes from the generalised gamma distribution (see Supplementary Note), retaining the mean and the variance on a logarithmic scale and thus fixing the heritability, while varying the $\theta$ parameter of the generalised gamma distribution between 0 and 2 (five settings of $\theta = 0, 0.5, 1, 1.5, 2$ with $\theta = 1$ corresponding to a Weibull distribution). For these data sets, we also varied the censoring levels of 0%, 20% and 40% (see Supplementary Note). For each of the censoring and phenotypic distribution combinations, 25 replicate phenotypic data sets were created, giving a total of 375 data sets. The prior parameters for $\sigma_G^2$, $\alpha$ and $\mu$ were set the same way as described in Data Analysis Details.

To compare our approach with other available methods we analyzed each data set using different approaches: a Cox Lasso[13], a martingale residuals approach with single-marker ordinary least squares (OLS) regression[11], and martingale residuals with a Bayesian regression mixture model with a Dirac spike (BayesR)[19]. For each of the 25 simulation replicates, across the five generalised gamma $\theta$ parameters, we calculated the correlation between the simulated genetic values and a genetic predictor, created from the regression coefficients obtained from each approach, in an independent data set (same number of markers, same causal markers and same effect sizes, with $N = 1000$ individuals), with results shown in Fig. 1a). Secondly, for all four methods, we calculated precision-recall curves for the generalised gamma distributions $\theta \in \{0, 1, 2\}$ and censoring levels 0%, 20% and 40% (Fig. 1b, Supplementary Fig. 2). Bayesian models used 5 chains with 1100 iterations for each chain with a burn-in of 100 and no thinning. The BayesW model was estimated using 11 quadrature points. The hyperparameter for Cox Lasso was estimated using 5-fold cross-validation for each simulated data set separately.

In addition, for the BayesW model, we analysed each of the 375 data sets, using only a single mixture distribution set to a constant of 0.01, or two mixture distributions with constants 0.01, 0.001. We compared these model formulations by accessing the slope between true and estimated non-zero marker effects (Fig. 1c)

and the estimated heritability, across the range of generalised gamma distributions (Fig. 1d). For each of the 5 generalised gamma $\theta$ parameter settings, we also calculated the mean false discovery rate (FDR) levels across the 25 replicate simulations given fixed posterior inclusion probabilities (Fig. 1e) for both the single-mixture and two-mixture model formulations (Fig. 1f). Finally, we tested the impact of using a different number of quadrature points by running the model for the Weibull setting data sets. We varied the number of quadrature points from 3 to 25 across 5 simulation replicates, using two mixture distributions (0.001,0.01), and investigated the root mean square error (RMSE: estimated/true) for marker effect estimates within 5 top deciles of the simulated marker effect distribution (Supplementary Fig. 3b).

To test the impact of LD among the markers, we used UK Biobank chromosome 22 imputed genotype data ($M = 194,922$ markers, $N = 20,000$ randomly selected individuals, $p = 2000$ randomly selected causal markers, with heritability $h^2 = 0.5$) and we simulated the phenotypes from Weibull distribution, with 25 simulation replicates. We used this data to compare BayesW to the same other methods described above, by calculating the correlation of simulated genetic value and a genetic predictor in an independent data set (the same number of markers, same causal markers and same effect sizes, with $N = 4000$ individuals). We present these results in Supplementary Fig. 1a. In addition, we used the same genetic data set but varied the censoring levels (Supplementary Fig. 1b), to examine the stability of the heritability estimate. Bayesian analyses used 5 chains with 3000 iterations each and a burn-in of 1000 and thinning of 5. The Cox Lasso model was trained the same way as in the uncorrelated case.

To validate properties of polygenicity, variance partitioning between mixtures and false discovery rate we used UK Biobank chromosome 1 imputed genotype data that was LD pruned with threshold $r^2 = 0.9$ as this data set was later used in the final analyses ($M = 230,227$ markers, $N = 25,000$ randomly selected individuals). We ran 10 simulations with three different numbers of causal loci: 200, 2500 and 4000. The phenotypes were simulated from Weibull distribution with a fixed heritability of $h^2 = 0.5$. All the models were executed with three variance components (0.0001,0.001,0.01) (Supplementary Figs. 11, 12). The effects were created by first grouping the markers via a clumping procedure (window size 10Mb, LD threshold $r^2 = 0.1$) and then assigning the effects to the index SNPs of randomly selected clumps.

Finally, we ran 10 simulations on the same UK Biobank chromosome 1 data as described in the previous section to check the performance of the BSP Gibbs sampling algorithm in a scenario that would be the closest to the empirical UK Biobank data analysis. Here, we only used $p = 2,500$ randomly selected causal SNPs, with heritability $h^2 = 0.5$. The phenotypes were simulated from Weibull distribution and models were run with three variance components (0.0001,0.001,0.01). Models were run by varying the number of tasks (parallelism) between 1, 4, 8, 16 and synchronisation rate (number of markers processed by each task until synchronisation) between 1, 5, 10, 20, 50 (Supplementary Fig. 3a). The scenario of 8 tasks (~30,000 markers per task) and synchronisation rate of 10 is used in the empirical data analysis.

**UK biobank data.** We restricted our discovery analysis of the UK Biobank to a sample of European-ancestry individuals ($N = 456,426$). To infer ancestry, 488,377 genotyped participants were projected onto the first two genotypic principal components (PC) in 2504 individuals of the 1000 Genomes project with known ancestries. Using the obtained PC loadings, we then assigned each participant to the closest population in the 1000 Genomes data: European, African, East-Asian, South-Asian or Admixed. As we wished to contrast the genetic basis of different phenotypes, we then removed closely related individuals as identified in the UK Biobank data release. While we expect that our model can accommodate related-ness similar to other mixed linear model approaches, we wished to compare phenotypes at markers that enter the model due to LD with underlying causal variants, and relatedness leads to the addition of markers within the model to capture the phenotypic covariance of closely related individuals.

We used the imputed autosomal genotype data of the UK Biobank provided as part of the data release. For each individual, we used the genotype probabilities to hard-call the genotypes for variants with an imputation quality score above 0.3. The hard-call-threshold was 0.1, setting the genotypes with probability ≤0.9 as missing. From the good quality markers (with missingness less than 5% and $p$-value for Hardy-Weinberg test larger than $10^{-6}$, as determined in the set of unrelated Europeans) were selected those with minor allele frequency (MAF) 0.0025 and rs identifier, in the set of European-ancestry participants, providing a data set of 9,144,511 SNPs, short indels and large structural variants. From this, we took the overlap with the Estonian Biobank data to give a final set of 8,433,421 markers. From the UK Biobank European data set, samples were excluded if in the UK Biobank quality control procedures they (i) were identified as extreme heterozygosity or missing genotype outliers; (ii) had a genetically inferred gender that did not match the self-reported gender; (iii) were identified to have putative sex chromosome aneuploidy; (iv) were excluded from kinship inference. Information on individuals who had withdrawn their consent for their data to be used was also removed. These filters resulted in a dataset with 382,466 individuals. We then excluded markers of high LD by conducting LD pruning using a threshold of $r^2 = 0.9$ for a 100kb window leaving us with a final set of 2,975,268 markers. This was done in order to decrease the number of markers that were in high LD and

thus giving very little extra information but requiring more than two times the computational resources. Genotype quality control was conducted using plink version 1.9[38].

We then selected the recorded measures for the 382,466 to create the phenotypic data sets for age-at-menopause, age-at-menarche and age-at-diagnosis of HBP, T2D or CAD. For each individual $i$ we created a pair of last known time (logarithmed) without an event $Y_i$ and censoring indicator $C_i$ ($C_i = 1$ if the person had the event at the end of the time period, otherwise $C_i = 0$). If the event had not happened for an individual, then the last time without having the event was defined as the last date of assessment centre visit minus date of birth (only month and year are known, the exact date was imputed to 15).

For age-at-menopause, we used UKB field 3581 to obtain the time if available. We excluded from the analysis (1) women who had reported having and later not having had menopause or vice versa, (2) women who said they had menopause but there are no record of the time of menopause (UKB field 2724), (3) women who have had a hysterectomy or the information about this is missing (UKB field 3591), 4) women whose menopause is before age 33 or after 65. This left us with a total of $N = 151,472$ women of which 108, 120 had the event and 43, 352 had not had an event by the end of the follow-up. For time-to-menarche we used UKB field 2714 and we excluded all women who had no record for time-to-menarche which left us with a total of $N = 200,493$ women of which all had had the event. For the age of diagnosis of HBP we used the UKB field 2966 for and we left out individuals who had the HBP diagnosed but there was no information about the age of diagnosis (UKB field 6150) which left us with a total of $N = 371,878$ individuals of which 95,123 had the event and 276, 755 had not had an event by the end of the follow-up. For age of diagnosis of T2D, we used either the UKB field 2976 or field 20009 or the mean of both two if both were available. We excluded individuals who had indicated self-reported "type 1 diabetes" (code 1222) and had Type 1 Diabetes (ICD code E10) diagnosis; we also excluded individuals who did not have any recorded time for the diagnosis of T2D but had indicated secondary diagnosis (UKB fields 41202 and 41204) of "non-insulin-dependent diabetes mellitus" (ICD 10 code E11) or self-reported non-cancer illness (UKB field 20002) "type 2 diabetes" (code 1223) or "diabetes" (code 1220). That left us with a total of $N = 372, 280$ individuals of which 15, 813 had the event and 356, 467 had not had an event by the end of the follow-up. For the age of diagnosis of CAD, we used either the minimum of age at angina diagnosed and age heart attack diagnosed (UKB fields 3627 and 3894) or the minimum age indicated to have either two of diagnoses (codes 1074, 1075) in UKB field 20009 or the mean of those if both were available. We excluded individuals who did not have any information about the time of diagnosis but had the following primary or secondary diagnoses: ICD 10 codes I20, I21, I22, I23, I24 or I25; self-reported angina (code 1074) or self-reported heart attack/myocardial infarction (code 1075). That left us with a total of $N = 360, 715$ individuals of which 17, 452 had the event and 343, 263 had not had an event by the end of the follow-up.

In the analysis we included covariates of sex, UK Biobank recruitment center, genotype batch and 20 first principal components of the LD clumped set of 1.2 million marker data set, calculated using flashPCA2[39] commit version b8044f1, to account for the population stratification in a standard way. We did not include any covariates of age or year of birth because these are directly associated with our phenotypes.

**Estonian biobank data.** The Estonian Biobank cohort is a volunteer-based sample of the Estonian resident adult population. The current number of participants-close to 52,000–represents a large proportion, 5%, of the Estonian adult population, making it ideally suited to population-based studies[40]. For the Estonian Biobank Data, 48,088 individuals were genotyped on Illumina Global Screening (GSA) ($N = 32,594$), OmniExpress ($N = 8102$), CoreExome ($N = 4,903$) and Hap370CNV ($N = 2,489$) arrays. We selected only those from the GSA array and imputed the data set to an Estonian reference, created from the whole genome sequence data of 2244 participants[41]. From 11,130,313 markers with imputation quality score >0.3, we selected SNPs that overlapped with the UK Biobank LD pruned data set, resulting in a set of 2,975,268 markers. The phenotypic data was constructed similarly to the phenotypes based on the UK Biobank data for the $N_{Est} = 32,594$ individuals genotyped on the GSA array. For time-to-event traits, if no event had happened then the time is considered censored and the last known age without the event was used, calculated as the last known date without event minus the date of birth. Because only the year of birth is known, birth date and month were imputed as July 1 for age calculations.

For age-at-menopause, we excluded women who had reported having menstruation stopped for other reasons which resulted in 6434 women who had had menopause and 12,934 women who had not had menopause. For age-at-menarche we excluded women who had not reported the age when the menstruation started which resulted in 18,134 women. For both age-at-menarche and age-at-menopause, if the event had occurred, self-reported age during that event was used.

Initially, the cases of CAD, HBP or T2D were identified on the basis of the baseline data collected during the recruitment, where the disease information was either retrieved from medical records or self-reported by the participant. Then, the information was linked with additional health insurance information that provided additional information on prevalent cases. To construct the phenotypes for the

time-to-diagnosis of CAD, HBP or T2D for the individuals with the corresponding diagnosis we used the age at the first appearance of the respective ICD 10 code that was also used for creating the UK Biobank phenotypes. If the self-reported data about the ICD 10 code has only the information about the year, the date and month were imputed as July 1 and if only the date is missing then the date was imputed as 15. Respective case-control phenotypes for CAD, HBP or T2D were defined 0 if the person had not had an event (censored) and 1 if the person had had an event and these binary indicators were adjusted for age and sex. For the T2D phenotype, we excluded individuals with a diagnosis of T1D. For CAD we resulted with 30,015 individuals without the diagnosis and 2579 individuals with a diagnosis, for HBP we resulted with 24,135 individuals without the diagnosis and 8459 individuals with a diagnosis and for T2D we resulted with 30,883 individuals without the diagnosis and 1457 individuals with a diagnosis.

**Data analysis details.** The BayesW model was run on the UK Biobank data without groups and with 20 MAF-LD groups that were defined as MAF quintiles and then quartiles within each of those MAF, quintiles split by the LD score. The cut-off points for creating the MAF quintiles were 0.006, 0.013, 0.039, 0.172; the cut-off points for creating LD score quartiles were 2.11, 3.08, 4.51 for the first; 3.20, 4.71, 6.84 for the second; 4.70, 6.89, 9.94 for the third; 7.65, 11.01, 15.70 for the fourth and 10.75, 15.10, 21.14 for the fifth MAF quintile. The prior distributions for the hyperparameters were specified such that they would be only weakly informative: normal priors would have a zero mean and very large variance, Dirichlet priors would be vectors of ones and the rest such that the prior parameter value would have a very small contribution to the conditional distribution compared to the likelihood. Specifically, for $\mu$ and $\delta$ the mean is chosen 0 and variance $\sigma_\mu^2 = \sigma_\delta^2 = 100$; for $\alpha$ we choose $\alpha_0 = 0.01$ and $\kappa_0 = 0.01$; for $\sigma_G^2$ in without groups and $\sigma_{G\varphi}^2, \forall \varphi$ in with groups model, we set parameters to be $\alpha_\sigma = 1, \beta_\sigma = 0.0001$; for $\pi$ and $\pi^\varphi$ the prior parameters is set to be a vector of ones. The model without groups was executed with mixture components 0.00001, 0.0001, 0.001, 0.01 (reflecting that the markers allocated into those mixtures explain the magnitude of 0.001%, 0.01%, 0.1% or 1% of the total genetic variance), and the model with groups was executed with (group-specific) mixture components (0.0001, 0.001, 0.01, 0.1). Guided by our simulation study (Supplementary Fig. 3b), we used 25 quadrature points for running each of the models. For the model without groups, we used five chains and for the model, with groups, we used three chains. Each of the chains was run for 10,000 iterations with a thinning of 5 giving us 2000 samples. We applied a stringent criterion of removing the first half of the chain as burn-in, giving the convergence statistics of Supplementary Figs. 4, 5, 6, 7. That gave 5000 samples for the model without groups and 3000 samples for the model with groups for each of the five traits.

The BSP Gibbs sampling scheme is implemented by partitioning the markers in equal size chunks assigned to workers (MPI tasks) themselves distributed over compute nodes. For the analyses we used 8 tasks per node; due to the differences in sample size we were using the different number of nodes to accommodate the data in memory: for time-to-menopause, we used 8 nodes, for time-to-menarche we used 10 nodes and for time-to-diagnosis of CAD, HBP and T2D we used 12 nodes. This resulted in splitting the markers between 64 workers for time-to-menopause, 80 workers for time-to-menarche and 96 workers for time-to-diagnosis of CAD, HBP or T2D. For the last case, the average number of markers assigned to one worker is 30,992. We chose to use a synchronisation rate of 10 meaning the synchronisation between all of the workers was done after sampling 10 markers in each of the workers. Both the choice of the maximum number of workers and the synchronisation rate are stringent options considering our simulation study results plotted in Supplementary Fig. 3a.

For testing region-based significance for BayesW, we used a Posterior Probability of the Window Variance (PPWV)[21]. PPWV requires first setting a threshold of the proportion of the genetic variance explained. Then, based on the posterior distributions we calculated the probability that each region explained more than the specified threshold of the proportion of the genetic variance and this quantity is denoted as PPWV. The regions were defined via the LD clumping procedure (window size 10 Mb, LD threshold $r^2 = 0.1$) resulting in regions that have high inter-region correlations but low intra-region correlations. For these LD clumped regions we used thresholds of 1/100,000, 1/10,000 and 1/1000 of the total genetic variance. The smallest threshold for PPWV is 1/100,000 of the total genetic variance as this gives the same magnitude as the smallest mixture component (0.00001) used in the models. The smallest mixture component reflects the smallest effect size the model is intended to capture. The thresholds of 1/10,000 and 1/1000 of the total genetic variance are chosen 10 and 100 times greater than the smallest threshold to point out the regions with larger effect sizes. To check the significance of the gene-associated regions we used more stringent thresholds of 1/10,000 and 1/1000 of the total genetic variance as gene-associated regions can contain greatly more markers. Furthermore, to make gene-associated regions more comparable, we fixed an upper bound of 250 for the markers that can contribute to a gene-associated, markers exceeding the bound were randomly discarded.

To do the comparison in terms of discovered regions and prediction accuracy we used the summary statistics from the fastGWA method[28]. Because there were no results for our definition of time-to-CAD or time-to-T2D we used time-to-

angina and time-to-heart attack summary statistics for comparison with CAD and time-to-diabetes for comparison with T2D. We called an LD clumped region significant if the region contained at least one SNP with a p-value < $5 \times 10^{-8}$. To do the prediction into the Estonian Biobank we only used the markers with p-value < $5 \times 10^{-8}$. We did the predictions only for age-at-menarche and age-at-menopause since the number of significant markers for them is higher.

To do the comparison in terms of predictive accuracy with a competing method we also trained the Cox-LASSO method with R package snpnet[14,15] with UK Biobank data and then used the estimates to make predictions into Estonian Biobank. To make the two models comparable, we used exactly the same data sizes for estimating the models on the UK Biobank as were used with the BayesW. For all of the traits, we decided to use 95% of the sample size as the training data and the rest as the validation data. This was done in order to minimise the loss in power due to not using the entire sample and 5% of the sample gives a sufficiently large validation set. We ran the Cox-LASSO model using snpnet with 16 threads and allocating 250 GB of memory. This was sufficient to find the optimal hyperparameter for the traits of time-to-menopause (22 iterations to find the optimal hyperparameter) and time-to-CAD (21 iterations to find the optimal hyperparameter). However, for the other traits, the snpnet procedure ran out of memory and it was decided to use the results from the last available iteration (iteration 28 for time-to-HBP, iteration 35 for time-to-menarche, iteration 27 for time-to-T2D). For the traits for which it was not possible to detect the optimal hyperparameter a sensitivity analysis was done by comparing with the previous iterations. Prediction accuracy was virtually the same between the last available iteration and some iterations before that suggesting that the last available iteration was already providing a hyperparameter close to the optimum.

The prediction based on BayesW into Estonian Biobank $\hat{g}$ was calculated by multiplying $\hat{g} = X_{Est}\hat{\beta}$, where $X_{Est}$ is $N_{Est} \times M$ matrix of standardised Estonian genotypes (each column is standardised using the mean and the standard deviation of the Estonian data), $\hat{\beta}$ is the $M \times I$ matrix containing the posterior distributions for $M$ marker effect sizes across $I$ iterations. To calculate the prediction into Estonia we used the BayesW model with groups using 3000 iterations which gave us posterior predictive distributions of the genetic values with 3000 iterations. To create the final predictor, we calculated the mean genetic value for each individual across 3000 iterations. We also created the predictor using the estimates from Cox-LASSO by multiplying the standardised Estonian genotype matrix with the vector of Cox-LASSO effect size estimates. We evaluated the performance of the two predictors by comparing them to the true phenotype value and calculating $R^2$ and Harrell's C-statistic[29]. Instead of using the exact phenotypes the martingale residuals from the Cox PH model where the true phenotype was regressed on the gender (if applicable) were used to calculate the $R^2$. That enables calculating the $R^2$ value using also the censored individuals. Harrell's C-statistic was calculated from the Cox PH model where the true phenotype was regressed on the predictor and gender (if applicable).

The BayesW calculations have been performed using the facilities of the Scientific IT and Application Support Center of EPFL and the Helvetios cluster. All of the post-analysis steps were conducted using R software (version 3.6.1)[42].

## Algorithm 1

Serial algorithm for BayesW sampling from the posterior distribution $p(\mu, \alpha, \delta, \gamma, \beta, \pi, \sigma_G^2 | D)$. Initialisation and prior specification.
**Data**: Matrix $x$ of standardised genotypes, matrix with covariate data $z$, vector of last time without an event $y$, vector of failure indicators $d$, prior hyperparameters $\alpha_0, \kappa_0, \alpha_\sigma, \beta_\sigma, \sigma_\mu^2, \sigma_\delta^2, p_L$, iterations $I$. $V_0^j$, $V_1^j$ and $V_2^j$ denote the partial sums of the exponentiated residuals (defined in Derivations for the sparse calculations), $K$ is the Euler-Mascheroni constant.
**Initial values**: Initially, we exclude all the variables from the model, thus the initial $\gamma_j = 0, \forall j \in \{1, \ldots, M\}$. We set $\beta_j = 0, \forall j$ and $\delta_q = 0, \forall q$. The initial value for $\alpha$ is chosen to be the suitably transformed variance of the log sample. The initial value for $\mu$ is the mean of the log sample. The $\sigma_G^2$ is initialized as the variance of the log sample divided by the total number of markers $M$.
**Parameters for prior distributions**: We set priors weakly informative. Otherwise, if available, prior information could be used. To get weakly informative priors, for $\alpha$ prior, we set parameters to be $\alpha_0 = 0.01$ and $\kappa_0 = 0.01$; for $\sigma_G^2$, we set parameters to be $\alpha_\sigma = 1$, $\beta_\sigma = 0.0001$; for $\mu$ prior, we set parameter $\sigma_\mu^2 = 100$ and similarly for $\delta_q$ we set parameter $\sigma_\delta^2 = 100$. The choice of prior parameters for $\pi$, $p_L$ is a vector of ones.

1 Initialise for every $i$: $\varepsilon_i = y_i - \mu$
2 **for** $iteration \leftarrow 1$ **to** $I$ **do**
3 Add the previous effect to the residual: $\varepsilon_i \leftarrow \varepsilon_i + \mu^{old}$
4 Sample $\mu$ using ARS;
5 Subtract the new effect from the residual: $\varepsilon_i \leftarrow \varepsilon_i - \mu$
6 Shuffle (covariates);
7 **foreach** covariate $q$ **do**
8 $\varepsilon_i \leftarrow \varepsilon_i + z_{iq}\delta_q^{old}$
9 Sample $\delta_q$ using ARS
10 $\varepsilon_i \leftarrow \varepsilon_i - z_{iq}\delta_q$
11 Sample $\alpha$ using ARS;
12 Shuffle (markers);
13 Calculate exponentiated residuals: $\epsilon_i \leftarrow \exp(\alpha\varepsilon_i - K)$;

14 **foreach** marker $j$ **do**
15 **if** $\beta_j^{old} = 0$ **then**
16 Calculate $V_0^j$, $V_1^j$, $V_2^j$;
17 **if** $\beta_j^{old} \neq 0$ **then**
18 $\varepsilon_i \leftarrow \varepsilon_i + x_{ij}\beta_j^{old}$;
19 $\epsilon_i \leftarrow \exp(\alpha\varepsilon_i - K)$;
20 Calculate $V_0^j$, $V_1^j$, $V_2^j$;
21 Sample mixture indicator $\gamma_j$;
22 **if** $\gamma_j > 0$ **then**
23 sample $\beta_j$ from the $\gamma_j$th conditional distribution using ARS;
24 $\varepsilon_i \leftarrow \varepsilon_i - x_{ij}\beta_j$;
25 $\epsilon_i \leftarrow \exp(\alpha\varepsilon_i - K)$;
26 **if** $\gamma_j = 0$ **then**
27 set $\beta_j = 0$;
28 Sample $\pi$;
29 Sample $\sigma_G^2$.

## Algorithm 2

Bulk Synchronous Parallel Gibbs sampling with BayesW. Data, parameter initialisation and prior values are set as in Algorithm 29.
**Input**: Define $T$ parallel workers (tasks) and synchronisation rate $u$. Each worker $t \in \{1, \ldots, T\}$ has its corresponding vector of marker effects $\beta^t$ (with the number of markers in each $\sim \frac{M}{T}$, $M$ is the total number of markers), indicator values $\gamma^t$ to update and set of $T$ messages $\Delta\varepsilon_t$: $N \times 1$, $N$ is the sample size.

1 Initialise variables;
2 **for** $iteration \leftarrow 1$ **to** $I$ **do**
3 Update $\mu, \delta, \alpha$ (as in Algorithm 29);
4 **foreach** subset of size $u$ **do**
5 $\Delta\varepsilon_t \leftarrow 0$, $t \in \{1, \ldots, T\}$;
6 **for** $t \leftarrow 1$ **to** $T$ in parallel **do**
7 **foreach** column $j$ from a subset of size $u$ of the columns assigned to worker $t$ **do**
8 **if** $\beta_j^{old} = 0$ **then**
9 Calculate $V_0^j$, $V_1^j$, $V_2^j$ based on $\epsilon$;
10 **if** $\beta_j^{old} \neq 0$ **then**
11 $\tilde{\varepsilon}_i \leftarrow \varepsilon_i + x_{ij}\beta_j^{old}$;
12 $\tilde{\epsilon}_i \leftarrow \exp(\alpha\tilde{\varepsilon}_i - K)$;
13 Calculate $V_0^j$, $V_1^j$, $V_2^j$ based on $\tilde{\epsilon}$;
14 Sample mixture indicator $\gamma_j$;
15 **if** $\gamma_j > 0$ **then**
16 sample $\beta_j$ from the $\gamma_j$th conditional distribution using ARS;
17 **if** $\gamma_j = 0$ **then**
18 set $\beta_j = 0$;
19 **if** $\left(\beta_j - \beta_j^{old}\right) \neq 0$ **then**
20 $\Delta\varepsilon_{ti} \leftarrow \Delta\varepsilon_{ti} - x_{ij}\left(\beta_j - \beta_j^{old}\right)$;
21 Wait until all workers are finished processing their sets of $u$ markers;
22 $\varepsilon_i \leftarrow \varepsilon_i + \sum_{t=1}^{T} \Delta\varepsilon_{ti}$;
23 $\epsilon_i \leftarrow \exp(\alpha\varepsilon_i - K)$;
24 Update $\pi$, $\sigma_G^2$ (as in Algorithm 29);

**Reporting summary**. Further information on research design is available in the Nature Research Reporting Summary linked to this article.

## Data availability

This project uses UK Biobank data under project 35520. UK Biobank genotypic and phenotypic data is available through a formal request at (http://www.ukbiobank.ac.uk). The UK Biobank has ethics approval from the North West Multi-centre Research Ethics Committee (MREC). For access to be granted to the Estonian Biobank genotypic and corresponding phenotypic data, a preliminary application must be presented to the oversight committee, who must first approve the project, ethics permission must then be obtained from the Estonian Committee on Bioethics and Human Research, and finally, a full project must be submitted and approved by the Estonian Biobank. This project was granted ethics approval by the Estonian Committee on Bioethics and Human Research (https://genomics.ut.ee/en/biobank.ee/data-access). The summary statistics for fastGWA method were accessed through http://fastgwa.info/ukbimp/phenotypes. Summaries of all posterior distributions obtained and full posterior distributions of the SNP marker effects sizes for each trait are deposited on Dryad (https://doi.org/10.5061/dryad.qbzkh18gp)[43]. Source data are provided with this paper.

## Code availability

Our BayesW model is implemented within the software Hydra, with fully open source code available at https://github.com/medical-genomics-group/hydra[44]. FlashPCA2 commit b8044f1 is available at https://github.com/gabraham/flashpca. fastGWA is part of the GCTA software, version 1.93.2 available at https://cnsgenomics.com/software/. snpnet commit version c6bc103 is available at https://github.com/junyangq/snpnet. plink version 1.9 is available at https://www.cog-genomics.org/plink/. R version 3.6.1 is available at https://www.r-project.org/.

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

## Acknowledgements

This project was funded by an SNSF Eccellenza Grant to MRR (PCEGP3-181181), and by core funding from the Institute of Science and Technology Austria and the University of Lausanne; the work of KF was supported by the grant PUT1665 by the Estonian Research Council. We would like to thank Mike Goddard for comments which greatly improved the work, the participants of the cohort studies and the Ecole Polytechnique Federal Lausanne (EPFL) SCITAS for their excellent compute resources, their generosity with their time and the kindness of their support.

## Author contributions

M.R.R. conceived and designed the study. M.R.R. and S.E.O. designed the study with contributions from A.K. and D.T.B. S.E.O., M.R.R., A.K., M.P. and D.T.B. contributed to the analysis. S.E.O., M.R.R., A.K. and D.T.B. derived the equations and the algorithm. S. E.O., E.J.O., D.T.B. coded the algorithm with contributions from AK and MRR. SEO and MRR wrote the paper. Z.K., K.L., K.F., and R.M. provided study oversight, contributed data and ran computer code for the analysis. All authors approved the final manuscript prior to submission.

## Competing interests

The authors declare no competing interests.
