## [Peer Review File · Nature Communications]

REVIEWER COMMENTS

Reviewer #3 (Remarks to the Author):

The authors developed a new Bayesian model, BayesW, to analyzing age-of-onset phenotypes vs. genetic markers. The authors show the value of their method in real data, perform extensive comparisons, and provide publicly available software. Based on their novel Bayesian model, they provide polygenicity of 5 complex traits comparing age of T2D, age of CAD, and age of HBP with age of menarche and menopause. Addressing some limitations and performing some additional analyses will strengthen the manuscript to help convince readers of the method and results.

1. The authors show evidence for polygenicity of age-of-onset for 5 complex traits. What would be compelling is if they also took a trait where only a few loci contribute to variance explained and show that their model did not see a contribution of thousands of common genetic variants.
2. The authors make claims on power but I disagree with using power for a Bayesian model and recommend not claiming power but improved model performance throughout.
3. On page 4, lines 157-162, the authors describing Figure 2a, describe age at menarche and age at CAD as highly polygenic while age at menopause and age at T2D are the least polygenic using proportions of variance explained in different mixture components. But all traits have the most variance explain in the 10-5 group. Should the comparison just be for the 10-3 mixture group?
4. Why were the thresholds of 1/1000, 1/10000, and 1/100000 of genotypic variance used for determining the posterior inclusion probabilities?
5. Sometimes the authors use snpnet and sometimes Cox-LASSO. It would be helpful if the authors were consistent in naming.
6. It is difficult and mostly inappropriate to compare a Bayesian model with a frequentist model (Table 1 results). The authors are suggesting that a PPWV of 0.95 is equivalent to a $p < 5e-8$ with no justification.
7. The authors show simulation results across a range of theta values in a generalized gamma distribution. In Figure 1a and c, when theta = 1, the phenotype model will be correctly specified, but in this case, the effect sizes, in general, shows less correlation with the true value and are underestimated, respectively. Why does a misspecified model appear to perform better for certain simulation results?
8. The approach assumes genotype data and converts dosages to hard-coded genotypes to make the method computational efficient. This will lead to a loss of information. Do SNPs with high imputation quality out-perform those of low imputation quality?
9. A table of descriptive statistics for the real data examples should be included,
10. The authors use 25 quadrature points for running the data analysis and justify this parameter based on the simulation study presented in Figure S3b, but that figure only shows up to quadrature points.

Reviewer #4 (Remarks to the Author):

This paper proposes a novel approach to joint analysis of GWAS data with time-to-event (censored survival) phenotypes that allows for joint analysis of all the SNPs rather than one at a time. The model involves mixture of normal distributions for SNP effect sizes that differ in their proportions of genetic variance explained (including one component with no effects), and the SNPs can be grouped a priori by such factors as minor allele frequency and LD score to provide insight into their distribution of effects. Posterior probabilities of inclusion are used to identify chromosomal regions harboring different kinds of effects. This is implemented in a Bayesian framework with an efficient hybrid sampling algorithm that is applicable to large-scale biobanks and illustrated on UK Biobank data for 5 age at diagnosis phenotypes.

The method is validated by extensive simulation studies, including comparisons with three other methods. Surprisingly, their survival analysis method provides roughly comparable performance to

their earlier methods for case-control phenotypes with double the sample size (where disease status is known but not the age at diagnosis). Although the method relies on a parametric Weibull distribution for the baseline hazard rather than the semi-parametric Cox regression, their simulations show robustness to misspecification within the larger family of generalized gamma distributions.

Interesting substantive findings about the genetic architecture of the various phenotypes, showing important differences between the reproductive traits of ages at menarche and menopause vs. the three chronic diseases (high blood pressure, cardiac disease, and type II diabetes), suggesting stronger evolutionary selection for the former. The UKB predictions are tested in the Estonian Genome Center data for these same phenotypes, showing greater prediction accuracy for their method compared with other methods.

The approach is explained in great detail in the Methods section, with a relatively simple description provided in the main text. I have no fundamental problems with the paper, other than a few very minor suggestions below:

Minor points

Lines 28-9: It's not clear why the martingale approach should have reduced power.

Line 84: should refer to Eq.(1), which is the same as Eq.(44) in the Supplement.

Line 107: it would be helpful to list the comparison methods here.

Line 110: "(of which Weibull is one)" – explain that this corresponds to $\theta=1$

Line 117: It would help if you briefly explain the concept of precision-recall curves and why you report these instead of the more familiar ROCs.

Line 129: BSG does not appear to be defined anywhere. Is this the same as your BSP Gibbs algorithm?

Lines 141-4: In the analyses with marker grouping, did you use the same four C^{ϕ}_k values for each MAF-LD group as in the ungrouped analysis?

Lines 166-8: The patterns in Figure 2 for age at menarche and age at menopause look quite different to me, with the latter seeming more like those for the late-onset diseases. What am I missing? That the age at menarche effect could have been under stronger evolutionary selection than the other traits (lines 170-1) is an interesting observation and seems quite reasonable; does that really apply also to age at menopause?

Lines 195-6: Although menarche and menopause have more concordance between the two methods than the other traits, it's worth mentioning that they have fewer hits than for fastGWA (since you emphasize the reverse for the other traits in the following sentence). The conclusions of this paragraph (lines 207-9) may be somewhat overstated.

Lines 235 ff: is Bayes R the same as BayesRR-RC defined on line 230? BayesR is not defined until line 501.

Line 343: Eqs.(8) and (9) are identical to (6) and (7), and (44). I don't see the point of repeating them. They're also the same as Eq.(1) in the main text, although it does make sense to have it both there and in the Supplement.

Lines 354-5: Does it make sense for the coefficients of covariate effects and the intercept to have

prior means of zero? Of course, this probably has little effect on the SNP effect estimates provided their prior variances are sufficiently large.

Line 359: Some guidance on the specification of the priors p_L for the π 's would be helpful (or at least how you specified them for your analyses).

Line 458: mention that the π 's should sum to 1.

Duncan C. Thomas

Reviewers' Comments: Genomic architecture and prediction of censored time-to-event phenotypes with a Bayesian genome-wide analysis

We thank the reviewers for their comments and the points that they raise, which clearly have the intention of improving the work we present, and we thank them for their time. We feel that the comments have been extremely helpful and our revisions have led to a greatly improved manuscript that is clearer in its purpose and presentation. We respond to each of the comments raised by the reviewers separately, including the line numbers in the manuscript where the changes have been made.

Reviewer 3 Remarks to the Author:

The authors developed a new Bayesian model, BayesW, to analyzing age-of-onset phenotypes vs. genetic markers. The authors show the value of their method in real data, perform extensive comparisons, and provide publicly available software. Based on their novel Bayesian model, they provide polygenicity of 5 complex traits comparing age of T2D, age of CAD, and age of HBP with age of menarche and menopause. Addressing some limitations and performing some additional analyses will strengthen the manuscript to help convince readers of the method and results.

We are very thankful to the reviewer for the comments. In order to provide an answer to the points 1 and 6 we conducted a simulation study to demonstrate the parameter estimates obtained from our model under different levels of polygenicity, explore the variance partitioning between mixtures and determine the false discovery rate. Furthermore, in the revised manuscript we further develop our PPWV approach defining genomic regions for discoveries based on the LD among markers. We show that this approach can more accurately represent the genetic architecture by combining markers in high LD, control the FDR, and provide improved model performance. We believe that these changes have made the conclusions clearer and more justified and we are grateful to the reviewer for their time.

1. *The authors show evidence for polygenicity of age-of-onset for 5 complex traits. What would be compelling is if they also took a trait where only a few loci contribute to variance explained and show that their model did not see a contribution of thousands of common genetic variants.*

We thank the reviewer for their concern. We believe that the only way to fully control polygenicity and to guarantee that a trait is affected only by a few loci is to simulate the data ourselves and then compare the results with the expectations. This is because before we run the analysis on real data, we would not know *a priori* what the underlying genetic architecture was. Therefore, to demonstrate that we would detect few loci if only a few underlying loci contribute toward the trait variance, we used real genomic data from the UK Biobank and simulated effects on top of the index SNPs from randomly picked LD clumps (clumps created using LD threshold of $r^2 = 0.1$, window size 10Mb) either with 200, 2500 or 4000 causal markers. For these simulations, we used UK Biobank chromosome 1 ($M = 230,227$, $N = 25,000$) and we simulated the phenotypes from Weibull distribution, heritability $h^2 = 0.5$. The results on polygenicity and effect size distribution are shown in Figure S11 and it is evident that if the number of causal loci is small (200, and thus the average effect size is high) then the model captures this information by including only a small number of markers with large effect sizes. Alternatively, if the number of causal markers is higher (4000, and thus the average effect size is small) then the model captures this information by including a large number of markers with small effect sizes. The corresponding results have been added in the manuscript in lines 148-152.

2. *The authors make claims on power but I disagree with using power for a Bayesian model and recommend not claiming power but improved model performance throughout.*

We wholeheartedly agree with the reviewer that we cannot make such claims on power for a Bayesian model. We have removed the parts that were advocating for higher power and replaced it emphasising better model performance instead.

3. *On page 4, lines 157-162, the authors describing Figure 2a, describe age at menarche and age at CAD as highly polygenic while age at menopause and age at T2D are the least polygenic using proportions of variance explained in different mixture components. But all traits have the most variance explain in the 10-5 group. Should the comparison just be for the 10-3 mixture group?*

We thank the reviewer for their comment. We agree that this description could be confusing and what we wished to say was the following: "Age-at-menopause and age-at-T2D diagnosis stand out with 32.3% (95% CI 28.9%, 35.7%) and 18.9% (95% CI 14.6%, 22.9%) of the genotypic variance attributable to the SNPs contributed by markers in the 10^{-3} mixture respectively (Figure 2b), indicating a substantial amount of genetic variance resulting from moderate to large effect sizes. In contrast, for the other traits the moderate to large effect sizes (mixture 10^{-3}) explain a far smaller part of the total genetic variance with age-at-menarche having almost no variance (0.1%, 95% CI 0.0%, 0.6 %) and only a small amount coming from that mixture for age-at-HBP diagnosis (5.6%, 95% CI 3.1%, 8.4%) and age-at-CAD (9.4%, 95% CI 6.5%, 12.9%)." We now state this addition in lines 190-197.

4. *Why were the thresholds of 1/1000, 1/10000, and 1/100000 of genotypic variance used for determining the posterior inclusion probabilities?*

The smallest threshold of 1/100,000 of the total genotypic variance was chosen to represent the same magnitude as the smallest mixture component in the ungrouped model (0.00001) or the smallest mixture component in the grouped model (0.0001 for each of the 20 groups). Smallest mixture component represents the smallest effect size we intend to capture with the model and therefore we have chosen 1/100,000 of the total genotypic variance to represent the smallest captured effect size. Other thresholds of 1/1000 and 1/10,000 of the total genotypic variance were chosen ten and hundred times higher compared to the smallest threshold to highlight the regions with greater effect sizes. We regret that the choice was not communicated clearly enough in the manuscript and we made the corresponding additions in lines 214-215 and more details are provided in lines 728-734.

5. *Sometimes the authors use `snpnet` and sometimes `Cox-LASSO`. It would be helpful if the authors were consistent in naming.*

We apologise that the naming was not consistent between `snpnet` and `Cox-LASSO` methods. We have now changed the manuscript such that the name of `Cox-LASSO` is used exclusively to represent the method and in the places where we use `snpnet` to estimate `Cox-LASSO` model, we highlight the fact of using the `snpnet` package.

6. *It is difficult and mostly inappropriate to compare a Bayesian model with a frequentist model (Table 1 results). The authors are suggesting that a PPWV of 0.95 is equivalent to a $p < 5e-8$ with no justification.*

We completely agree that it is very difficult to compare frequentist and Bayesian models and here, we do not expect to provide a comprehensive comparison between the two. However, we wish to validate the magnitude and concordance of the regions discovered by the two methods. Thanks to this comment, we realised that in the previous manuscript we did not include our justification for comparing PPWV and p-value based discoveries. Namely, to compare the number of discovered regions we need to verify that the discoveries contain only a limited number of false positives for both methods, in other words, type I error has to be controlled before examining properties of recall. It is well established that a p-value threshold of $5 \cdot 10^{-8}$ should control for false positives and to fill the gap for PPWV based discoveries for LD clumped regions we conducted simulations (simulation settings described in the answer to point 1) to see how does false discovery rate behave throughout a different number of causal markers. The results in Figure S12 demonstrate that using a PPWV threshold of 0.9 keeps the false discovery rate below 0.05. Therefore, we have used the PPWV threshold of 0.9 for BayesW when comparing the numbers of discovered LD clumped regions between fastGWA and BayesW. We believe that this should be sufficient to justify the comparisons in terms of magnitude and concordance between the two methods. We added the descriptions about the additional simulations in lines 155-158 and 581-588, further clarification was added in lines 233-250. Table 1 has been updated such that it is now using LD clumped regions instead.

7. *The authors show simulation results across a range of theta values in a generalized gamma distribution. In Figure 1a and c, when $\theta = 1$, the phenotype model will be correctly specified, but in this case, the effect sizes, in general, shows less correlation with the true value and are underestimated, respectively. Why does a misspecified model appear to perform better for certain simulation results?*

We thank the reviewer for pointing out the two interesting phenomena that might seem counter-intuitive. Firstly, in Figure 1a, we observe that the generalised gamma distributions with $\theta > 1$ lead to more accurate genetic predictions compared to the Weibull model ($\theta = 1$). This is because such phenotypic distributions are easier to discriminate meaning that for distributions where $\theta > 1$ the same difference in genetic values leads to greater phenotypic distribution differences in Kullback-Leibler divergences compared to $\theta = 1$. However, if we constructed a model specifically for the case if $\theta = 2$ then this kind of hypothetical model would likely outperform BayesW for this specific case. Therefore, the differences mostly reflect the property that some distributions are easier to discriminate. However, the key message of the figure remains the same as throughout different phenotypic distributions BayesW outperforms other available methods. The explanation is added in lines 122-126.

Secondly, in Figure 1c we observe a very slight underestimation of the effect sizes even if the model correctly specified. We believe that such slight underestimation is expected considering that using Gaussian priors for maximum *a posteriori*

corresponds to ridge regression estimates which is known for shrinking the effect size estimates. Hence, even the correctly specified model can lead to slight effect size underestimation due to using Gaussian priors. To examine the property more closely, we zoomed in to one random iteration used in Figure 1c (Figure S9) for $\theta = 0, 1, 2$. Visual inspection of Figure S9 shows no remarkable differences between the estimates of three phenotypic distributions, regressing true effect size on the effect size estimate (blue line) still shows a slight underestimate for the correctly specified model ($\theta = 1$) and no misestimation for $\theta = 0$. We believe this result is due to the inflation in the σ_G^2 hyperparameter estimates as shown in Figure 1d if $\theta = 0$. In return, increased σ_G^2 hyperparameters make the ridge regression regularisation term smaller leading to a smaller bias if $\theta = 0$. Thus, in general, we acknowledge the fact that using the Bayesian framework can lead to slight effect size underestimation and the misspecified model with inflated hyperparameters can have a smaller bias for effect size estimates compared to the correctly specified model. Nevertheless, we find such slight underestimation an inherent part of the Bayesian framework which still greatly improves the modelling and as seen in Figure S9 the effects of this underestimation seem almost negligible when comparing different estimates for different phenotypes. The explanation is added in lines 132-140.

8. *The approach assumes genotype data and converts dosages to hard-coded genotypes to make the method computational efficient. This will lead to a loss of information. Do SNPs with high imputation quality out-perform those of low imputation quality?*

Naturally, we believe that this is the case as high imputation quality SNPs will likely outperform SNPs with low imputation quality. SNPs with low imputation quality can be seen as those with a degree of error added to their values, which potentially leads to misclassification of the hard-coded genotypes. This would result in reduced covariance between the marker and the trait (we cannot see how SNPs with poor imputation quality could lead to the increased association as all imputation errors would have to align with a phenotypic value). We now mention this in the results as a limitation of our study in lines 360-366. However, we do not believe this to be a hindrance to our method or the application in this work. Hard-coded genotypic values will likely be the norm with the upcoming release of whole-genome sequence data and our aim is to provide a time-to-event model that is capable of scaling to these data requirements. Second, we apply our approach to markers that are imputed in both the UK Biobank and the Estonian genome centre data and by selecting markers present in both populations we are favouring markers that impute well across human populations.

9. *A table of descriptive statistics for the real data examples should be included,*

We agree with this comment. A table of descriptive statistics has been added to the manuscript as Table S1, summarising proportions of uncensored individuals, means and standard deviations of uncensored event times and range statistics.

10. *The authors use 25 quadrature points for running the data analysis and justify this parameter based on the simulation study presented in Figure S3b, but that figure only shows up to quadrature points.*

We thank the reviewer for pointing out this inconsistency. We conducted further simulations to demonstrate that using 25 quadrature points that was used in the real data analysis also yields similar accuracy compared to other numbers of quadrature points after having reached plateau (quadrature points ≥ 7). The updated results are shown in Figure S3. Besides, we point out that in general, using more quadrature points is always the preferred choice as it leads to a more accurate approximation of the integral.

Reviewer 4 Remarks to the Author:

This paper proposes a novel approach to joint analysis of GWAS data with time-to-event (censored survival) phenotypes that allows for joint analysis of all the SNPs rather than one at a time. The model involves mixture of normal distributions for SNP effect sizes that differ in their proportions of genetic variance explained (including one component with no effects), and the SNPs can be grouped a priori by such factors as minor allele frequency and LD score to provide insight into their distribution of effects. Posterior probabilities of inclusion are used to identify chromosomal regions harboring different kinds of effects. This is implemented in a Bayesian framework with an efficient hybrid sampling algorithm that is applicable to large-scale biobanks and illustrated on UK Biobank data for 5 age at diagnosis phenotypes.

The method is validated by extensive simulation studies, including comparisons with three other methods. Surprisingly, their survival analysis method provides roughly comparable performance to their earlier methods for case-control phenotypes with double the sample size (where disease status is known but not the age at diagnosis). Although the method relies on a parametric Weibull distribution for the baseline hazard rather than the semi-parametric Cox regression, their simulations show robustness to misspecification within the larger family of generalized gamma distributions.

Interesting substantive findings about the genetic architecture of the various phenotypes, showing important differences between the reproductive traits of ages at menarche and menopause vs. the three chronic diseases (high blood pressure, cardiac disease, and type II diabetes), suggesting stronger evolutionary selection for the former. The UKB predictions are tested in the Estonian Genome Center data for these same phenotypes, showing greater prediction accuracy for their method compared with other methods.

The approach is explained in great detail in the Methods section, with a relatively simple description provided in the main text. I have no fundamental problems with the paper, other than a few very minor suggestions below:

We are extremely grateful to the reviewer for the thorough and helpful feedback. Here, we provide the answers to the comments and issues highlighted by the reviewer.

1. *Lines 28-9: It's not clear why the martingale approach should have reduced power.*

Martingale residual approach combines the information about censoring indicator and phenotypic value into one summary statistic and the censoring information is not included in the model likelihood. Combining the information in one summary statistic can result in a loss of information. Methods such as BayesW or Cox LASSO are including the censoring information explicitly in the likelihood, and thus better capture the information from the censored individuals. We also cleared the explanation in the manuscript in lines 26-30.

2. *Line 84: should refer to Eq.(1), which is the same as Eq.(44) in the Supplement.*

We fully agree with the suggestion and we now refer to Eq.(1) instead of in the text.

3. *Line 107: it would be helpful to list the comparison methods here.*

We fully agree with the idea and we now list the methods used for the comparison simulations in lines 110-112.

4. *Line 110: "(of which Weibull is one)" – explain that this corresponds to $\theta=1$*

We agree with the suggestion and added the explanation in lines 115-116.

5. *Line 117: It would help if you briefly explain the concept of precision-recall curves and why you report these instead of the more familiar ROCs.*

We now added additional explanations about precision-recall curves and explained the reasons why we choose to use them. It has been shown that in the settings where the classes are greatly imbalanced, for example, the number of markers with an effect is greatly smaller than the number without an effect, it is more indicative to use precision-recall curves. Moreover, it has been shown that dominating precision-recall curves lead to dominating ROCs and therefore the final conclusions in terms of domination will be the same for both curves. The clarification has been added in lines 128-130.

6. *Line 129: BSG does not appear to be defined anywhere. Is this the same as your BSP Gibbs algorithm?*

We are very sorry for the inconsistency when using this abbreviation. We have now corrected the manuscript such that Bulk Synchronous Parallel is only abbreviated as BSP throughout the manuscript and incorrectly used BSG that was referring to "Bulk Synchronous Parallel Gibbs" is now replaced with "BSP Gibbs".

7. *Lines 141-4: In the analyses with marker grouping, did you use the same four C_k^ϕ values for each MAF-LD group as in the ungrouped analysis?*

In the analyses with marker grouping, we increased the size of the mixture constant values C_k^ϕ ten times compared to the ungrouped analysis and therefore all of the MAF-LD groups are using mixture constants (0.0001,0.001,0.01,0.1) such that they would represent 0.01%, 0.1%, 1% or 10% of the group-specific genetic variances. It was chosen to increase the mixture constants used in the grouped models as there the mixtures represent group-specific genetic variances which are smaller than the total genetic variance used in the ungrouped model. We also added this explanation to the manuscript in lines 172-174.

8. *Lines 166-8: The patterns in Figure 2 for age at menarche and age at menopause look quite different to me, with the latter seeming more like those for the late-onset diseases. What am I missing? That the age at menarche effect could have been under stronger evolutionary selection than the other traits (lines 170-1) is an interesting observation and seems quite reasonable; does that really apply also to age at menopause?*

We thank the reviewer for their comment and we agree that the conclusion for age-at-menopause can appear less pronounced compared to age-at-menarche. However, we believe that age-at-menarche and age-at-menopause still stand out as the patterns seen in figures 2c and 2d indicate that we find more of the genetic variance coming from the low MAF quintiles (1-3) for age-at-menarche and age-at-menopause compared to the other traits. To clarify the manuscript, we now state in the text the following section in lines 199-203.

"For age-at-menarche, many rare low-LD SNPs and many common SNPs contribute similar proportions to the phenotypic variance attributable to the SNP markers, implying larger absolute effect sizes for rare low-LD variants per minor allele substitution, with age-at-menopause showing a similar but less pronounced pattern with a noticeable proportion of the genetic variance stemming from small effect sizes of the rare variants (Figure 2d, MAF quintiles 1-3)."

9. *Lines 195-6: Although menarche and menopause have more concordance between the two methods than the other traits, it's worth mentioning that they have fewer hits than for fastGWA (since you emphasize the reverse for the other traits in the following sentence). The conclusions of this paragraph (lines 207-9) may be somewhat overstated.*

We thank the reviewer for their comment, which has led us to revise our PPWV approach (explanation added in lines 730-738) to incorporate LD among the markers leading to a more biologically motivated way for grouping and determining the probability that each LD block of correlated markers contributed at least 0.001% of the genetic variance. Now, the number of discovered regions for age-at-menarche and age-at-menopause are rather similar between BayesW and fastGWA with only slightly more regions discovered by fastGWA (433) compared to BayesW (414) for age-at-menarche. We now state the result of fewer hits in lines 243-244. Additionally, this approach led us to some interesting findings indicating that regions that are identified by BayesW and not by fastGWA have elevated fastGWA p-values and regions identified by fastGWA and not by BayesW have elevated PPWV values (Figure S13). We also rephrased the conclusion of the paragraph (lines 253-256) in the following way:

"Therefore, BayesW identifies already found regions along with novel regions compared to previous association methods; for time-to-diagnosis traits, it can discover more regions due to using the censored individuals; and BayesW results yield greatly improved prediction accuracy compared to fastGWA."

10. *Lines 235 ff: is Bayes R the same as BayesRR-RC defined on line 230? BayesR is not defined until line 501.*

We apologise for the inconsistency in the text. Throughout the manuscript we use the term BayesR to refer to the type of model (Bayesian mixture model for normally distributed phenotypes) and to make comparisons with BayesW model that assumes phenotype follows Weibull distribution. In this specific example (lines 273-292) BayesR type model was estimated using the latest extension for groups (the group approach is called BayesRR-RC). We have now changed the manuscript such that BayesR term is used exclusively and it is specified in addition if the BayesR type model was estimated using groups.

11. *Line 343: Eqs.(8) and (9) are identical to (6) and (7), and (44). I don't see the point of repeating them. They're also the same as Eq.(1) in the main text, although it does make sense to have it both there and in the Supplement.*

We agree with the comment and we have now removed the former equations (8) and (9). We retained the equation (1) so that it would be easier to follow the main text and also equation (42) (former equation 44) to emphasise the aspect of using grouped model, not the ungrouped model as shown in equation (6).

12. *Lines 354-5: Does it make sense for the coefficients of covariate effects and the intercept to have prior means of zero? Of course, this probably has little effect on the SNP effect estimates provided their prior variances are sufficiently large.*

Throughout this paper, the idea was to enforce weakly informative priors and that is the reason behind the choice of a prior mean of zero with a very large variance that is a common choice to create weakly informative normal priors. The prior variance is chosen to be very large ($\sigma^2 = 100$) to yield only a very weakly informative prior. The choice of using the priors is now explained further in lines 704-710.

13. *Line 359: Some guidance on the specification of the priors p_L for the π 's would be helpful (or at least how you specified them for your analyses).*

We choose to use weakly informative priors also for the prior inclusion probabilities (vectors π and $\pi_\phi, \forall \phi$). Therefore, we use a common prior choice of a vector of ones for all such cases. To highlight this decision we made the changes to the manuscript in lines 704-710.

14. *Line 458: mention that the π 's should sum to 1.*

We thank the reviewer for a useful comment, we now mention in line 512 that the elements in π are constrained to sum up to one.

REVIEWERS' COMMENTS

Reviewer #3 (Remarks to the Author):

I appreciate the thorough consideration and responses to my previous comments. I have no further comments.

Reviewer #4 (Remarks to the Author):

One of the most diligent responses to reviews I can recall seeing, especially the detail rebuttal letter in its clear cross-referencing to the manuscript. Well done! No further comments.

I leave it to the other reviewer to comment on the adequacy of the responses to their critique, but it looks good to me.